



# Exploring the driving factors of compound flood severity in coastal cities: a comprehensive analytical approach

Yan Liu[1], Ting Zhang[1], Yi Ding[2], Aiqing Kang[3], Xiaohui Lei[3], Jianzhu Li[1]

[1]State Key Laboratory of Hydraulic Engineering Intelligent Construction and Operation, Tianjin University, Tianjin 300072, China
[2]Faculty of Architecture, Civil And Transportation Engineering, Beijing University of Technology, Beijing 100124, China
[3]State Key Laboratory of Simulation and Regulation of Water Cycle in River Basin, China Institute of Water Resources and Hydropower Research, Beijing 100038, China

*Correspondence to*: Ting Zhang (zhangting_hydro@tju.edu.cn)

**Abstract.** Coastal cities frequently face various types of flooding triggered by heavy rainfall and storm surges, such as fluvial flooding and pluvial flooding. Currently, Currently, there is a lack of comprehensive methods to analyse the sources of severe compound flooding. This study, using the Shahe River Basin in Guangzhou, China as an example, establishes and validates a coupled 1D and 2D hydrodynamic model. Based on historical data, it constructs joint probability distributions of rainfall and tidal levels with different return periods and durations. Using the results from the coupled model under various design scenarios, it proposes an impact index to quantify the contributions of rainfall and tides to flooding. Furthermore, it quantifies the interactions between fluvial flooding and pluvial flooding. Flood-prone areas are delineated, and the causes of flooding are analyzed.The results show that when the return periods of rainfall and tide level are both 10 years, the Kendall return period for the combined event of rainfall and tide level is 36.35 years, greater than the "Or" return period (5.40 years) and less than the "And" return period (66.88 years). The impact degree index of rainfall on flooding varies between 0.5 and 1, with the minimum at 24-hour duration, indicating that the study area is primarily affected by rainfall and the influence of tide level is most significant at 24-hour duration. The pluvial flooding caused by the influence of river water level on the drainage outlet accounts for 19.08% of the total volume at most. This shows that fluvial flooding affects the seriousness of pluvial flooding by influencing the water levels of outlets. The flood-prone area is divided into different regions based on the main natural factors (rainfall and tidal level) and social factors (pipeline network, drainage outlets, and riverbank defenses) to help decision-makers identify the causes of flooding in each drainage unit and better formulate targeted disaster reduction strategies to improve flood control capabilities.

## 1 Introduction

According to the Intergovernmental Panel on Climate Change (IPCC) in its Sixth Assessment Report, released in 2021, it is stated that heavy rainfall and floods in many regions worldwide are expected to intensify and become more frequent in the 21st century (Masson-Delmotte et al., 2021). This trend has already become evident in various countries, posing a serious





threat to nations and people. For example, in this year, severe rainstorms in February led to significant casualties in the state of São Paulo, Brazil. In May, the Emilia-Romagna region in northern Italy was hit by heavy rainfall, resulting in at least 14 fatalities and 305 landslides. In July, both Pakistan and China experienced severe rainstorm and flood disasters. The direct cause of these disasters is heavy rainfall. When the intensity of rainfall is high and its duration is prolonged, urban surface

and underground drainage systems often struggle to handle such a large volume of water. Coastal areas are also impacted by storm surges, such as the southeastern coastal regions of China and the eastern coast of the United States. Storm surges cause sea level rise, which may inundate urban areas and affect floodgate discharge, further exacerbating urban flood risks.

To better understand the flood risks caused by compound events of heavy rainfall and storm surges, researches have focused on exploring the interdependence of various factors in the fields of hydrology, meteorology, and oceanography using Copula

theory (Pappadà et al., 2018; Zellou and Rahali, 2019). Wahl et al. analyzed the likelihood of compound events of storm surges and heavy rainfall occurring in coastal areas of the United States, and the results showed higher flood risk in the US east and Gulf coasts area (Wahl et al., 2015). Yang and Qian proposed using the Particle Swarm Optimization (PSO) algorithm to estimate the marginal cumulative distribution of wind speed, storm surges, and heavy rainfall, as well as the parameters of the three-variable joint function (Yang and Qian, 2019). Latif and Simonovic combined rainfall, storm surges,

and river discharge observations to create a three-variable probability framework (Latif and Simonovic, 2022).

Combining joint distribution models with return periods allows the design of multivariate combination scenarios under different return periods. Extensive researches have focused on identifying flood risks under various combinations of variables. Urban flood modeling techniques are powerful tools for supporting these studies (van Dijk et al., 2013). While modeling methods tend to evolve towards artificial intelligence algorithms, one-dimensional (1D) and two-dimensional (2D)

coupled numerical models with interpretable physical processes remain popular. Zhang et al. designed rainfall scenarios under different return periods and compared urban flooding results under different rainfall scenarios using MIKE FLOOD model (Zhang et al., 2022). Lian et al. evaluated the combined impact of rainfall and tide levels on flood risk in coastal cities and found that the greatest threat comes from heavy rainfall, with tide levels adding additional flood risk (Lian et al., 2013). However, these studies did not quantify the degree of impact of rainfall and storm surges on flooding. Lian et al. proposed a

method to divide the flood-prone area into three regions based on the magnitude of their impact, namely, rainfall area, tidal area, and common area (Lian et al., 2017). The common area is defined as the region where flooding is influenced by both tides and rainfall. Their study used daily rainfall and daily tide levels but did not consider the effect of duration. In urban environments, short-duration heavy rainfall events are more likely to occur, making it particularly important to study the differential effects of rainfall and tide levels on flooding under different durations.

Urban flooding encompasses both fluvial flooding, which results from inadequate river capacity, and pluvial flooding, which occurs due to inadequate drainage in urban infrastructure. In undeveloped areas, only fluvial flooding exists. It is evident that pluvial flooding emerges due to urbanization. Therefore, urban flooding is the result of the interaction between natural factors like rainfall and societal factors like urban drainage systems. Skougaard Kaspersen et al. compared the impact of climate change and urban development patterns on the exposure of four European cities to floods (Skougaard Kaspersen et



al., 2017). Pervin et al. analyzed the impact of improving drainage infrastructure and proper solid waste management on reducing urban flood risk (Pervin et al., 2019). Huang et al. categorized the causes of flooding in Guangzhou, China into two types, low-lying terrain and inadequate drainage facilities (Huang et al., 2018). These studies provide valuable insights for developing flood prevention and mitigation measures. However, they tend to combine fluvial flooding and pluvial flooding into a single category. As a result, there may be a lack of comprehensive understanding of the mechanisms of interaction

between fluvial flooding and pluvial flooding. Fluvial flooding and pluvial flooding have distinct definitions but are closely related and can transform under certain conditions. For example, if upstream efforts increase drainage capacity, it may lead to increased downstream flood flow, thereby increasing the risk of exacerbating downstream fluvial flood disasters. On the other hand, if the river maintains high water levels continuously, even without levee breaches, it may result in drainage difficulties due to floodwater overtopping the outfall. This can worsen pluvial flooding. There is few research on the

interactions between different types of flooding. This complexity and diversity make flood problems more challenging. To address flood problems, it is necessary to study the influence of natural factors on flooding and the interaction of different types of flooding.

The purpose of this study is to propose a universal method to enhance the understanding of the risk and interaction of different types of floods under the combined impact of rainstorms and the tide levels, and to identify the causes of urban

flooding. Section 2 introduces the study area, the Shahe River Basin in Guangzhou, China, and provides information of data availability. Section 3 constructs and validates the stormwater flood model, and briefly describes the concepts and mathematical formulas of bivariate joint distribution, impact degree index, and spatial interaction forces. Section 4 determines different combinations of scenarios with different rainstorms and tide levels, analyzes the degree of impact of different-duration rainstorms and tide levels on flooding, the interaction process between fluvial flooding and pluvial

flooding, and provides the causes of flooding in drainage units. Section 5 discusses the effects of rainfall and tide level process on flooding and the study trends of urban flood causes. Finally, Section 6 draws the conclusion. The study results can support the development of precise flood prevention and control for different types of floods.

## 2 Study area and data

### 2.1 Study area

The Shahe River Basin is located in the center of Guangzhou, China, with a total area of 36.77 km$^2$. The topography of the area slopes from north to south and from west to east, with elevations ranging from 7.31 to 35.0 m. In the entire area, 63% of the land is covered with hard surfaces, and the water surface occupies only 1.2% of the total area. The main river in the area is the Shahe River, which flows from north to south and has a main channel length of 14.24 km. The average slope of the main channel is 1.71‰. There is a mini reservoir named Pachili in the area, with a catchment area of 2.04 km$^2$ and a total

capacity of $1.44 \times 10^4$ m$^3$. At the mouth of the Shahe River, there is a tidal gate. No drainage pump station has been built at the mouth of the river, which creates significant drainage pressure during high tide periods. Downstream along Guangzhou





Avenue, there is a flood diversion channel, but no tidal gate is installed at the outlet, which poses a risk of tidal backflow. The geographical location is shown in Fig 1.

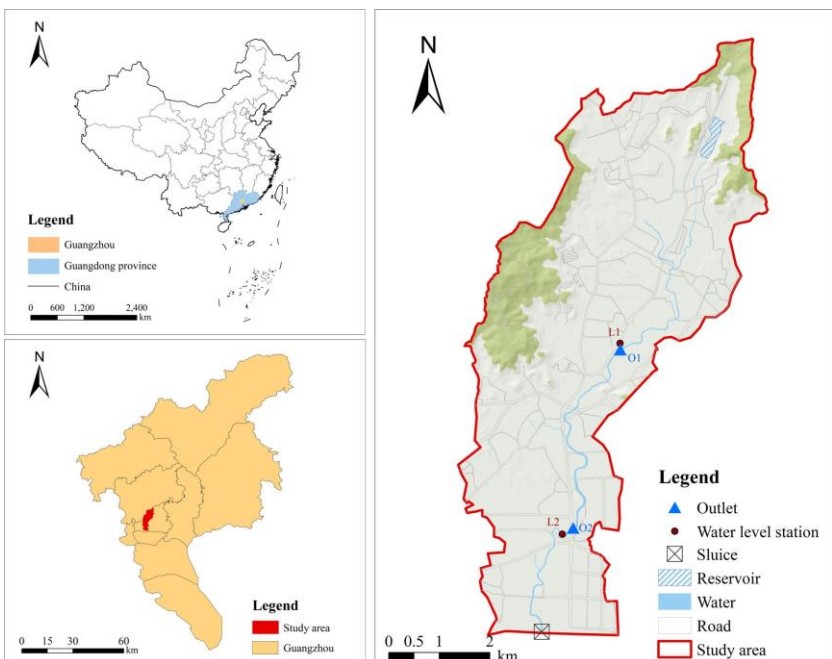

**Figure 1: Location of the study area.**

### 2.2 Data sources

This study utilized rainfall data at 15-minute intervals and tidal data at 5-minute intervals from 2006 to 2021 for rainfall-tide scenario design. The DEM with a spatial resolution of 5 m, land use data, drainage network system data, river section data, and hydraulic engineering parameters were used for constructing the urban flood model. Measured rainfall, water level data and water depth data on May 23, 2023, were used for model calibration. The topographic data were sourced from the Guangzhou Municipal Planning and Natural Resources Bureau, while other data were provided by the Guangzhou Water Authority.

## 3 Methods

### 3.1 Framework

In this study, Copula functions are used to construct combinations of rainfall and tidal scenarios with different durations. The Pilgrim & Cordery method and the same-frequency method are employed to design short-duration and long-duration rainfall





events respectively, while the modified equal-multiple method is used to design tidal processes. 1D-2D hydraulic coupling
model based on the topography data is established to simulate the flooding under different scenarios. Subsequently, the
impact degree index is introduced to quantify the impact of rainfall and tidal levels on flooding, and the differences in
various drainage units for different durations are analyzed. The study divides the flooding events into five stages, with key
time nodes being when the river water level exceeds the elevation of the drainage outlets and when it overflows the
riverbank. By examining the flooding volumes at different stages, this study illustrated the amplifying effects of the
interactions in flood events. Finally, the study summarized the causes of flooding. The research framework is depicted in Fig.
2.

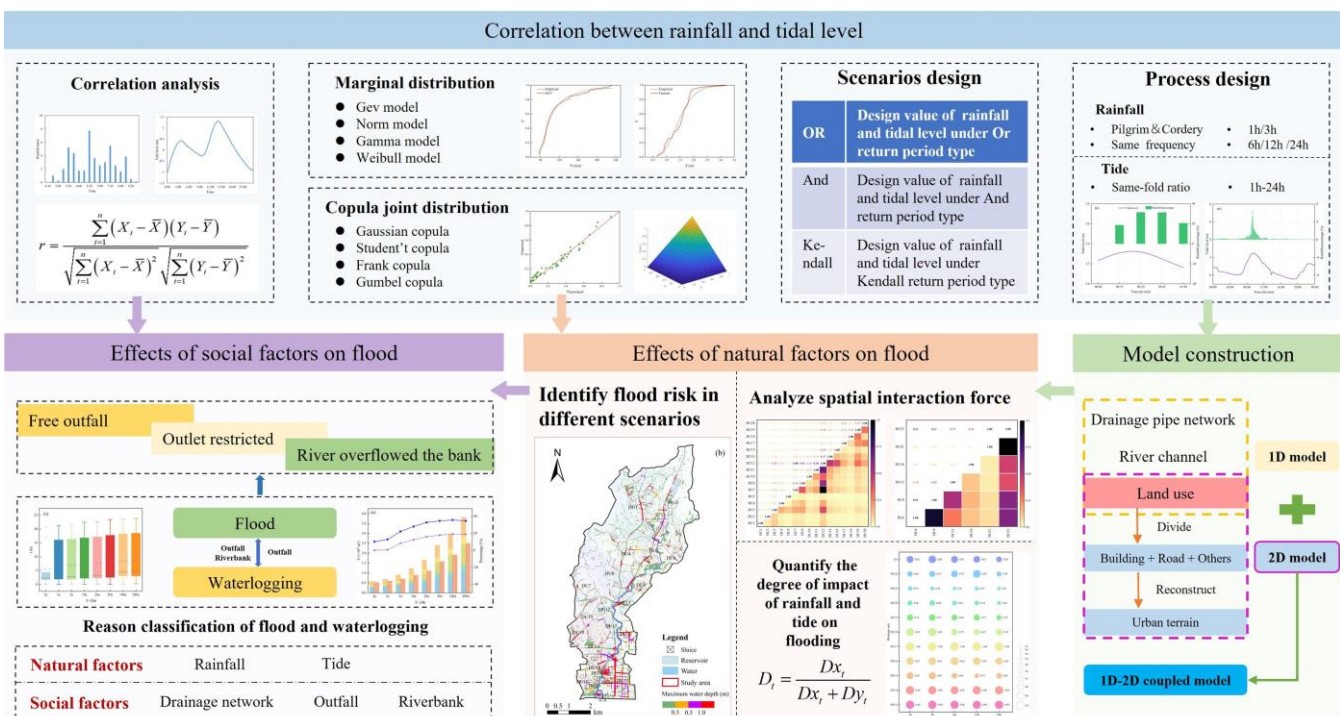

**Figure 2: Research framework diagram.**

## 3.2 Model construction and verification

### 3.2.1 Model construction

In this study, 1D-2D hydrodynamic coupling model is constructed to study the response of urban flooding to different
rainstorm and tidal level scenarios. The 1D hydraulic model consists of the urban drainage network model and the 1D river
model. It utilizes the Saint-Venant equations to perform calculations. The governing equations are expressed as follows:





$$\frac{\partial A}{\partial t} + \frac{\partial Q}{\partial x} = 0 \tag{1}$$

$$\frac{\partial Q}{\partial t} + \frac{\partial}{\partial X}\left(\frac{Q^2}{A}\right) + gA\left(\frac{\partial Z}{\partial X} + S_f\right) = 0 \tag{2}$$

where $x$ is the distance (m); $t$ is the time (s); $Q$ is the volumetric flow rate (m³/s); $A$ is the flow area (m²); $g$ is the acceleration of gravity (m/s²); and $Sf$ is the friction slope.

The 2D hydraulic model adopts the 2D shallow water equations, i.e. the depth-averaged Navier-Stokes equations, to mathematically describe the 2D flow dynamics. It assumes that water flow primarily occurs in the horizontal direction, while variations of flow velocity in the vertical direction are neglected. The storage cell method is used to solve the equation. For more details, our previous research results can be referenced (Wei et al., 2023). The governing equation can be represented as follows:

$$\frac{\partial \eta}{\partial t} + \frac{\partial \theta}{\partial X} + \frac{\partial \varphi}{\partial Y} + S_b + S_f = 0 \tag{3}$$

where $X$ and $Y$ are the spatial coordinates; $t$ is the time (s); $\theta$ and $\varphi$ are the convective flux variables in the $X$ and $Y$ directions, respectively; $S_b$ and $S_f$ represent the source term due to the slope of the terrain and the bed friction, respectively; and $\eta$ is the conservation variable.

To improve the representation of terrain data in road and the vertical abrupt changes, the study area is divided into three parts: roads, buildings, and others. Within the road area, elevation measurement points are used as references for nearest-neighbor interpolation. In the building area, vertical adjustments are made to the DEM data to reconstruct the terrain. This approach is based on the research by Huang et al (Huang et al., 2023).

### 3.2.2 Model verification

Based on different land use types, infiltration parameters for different catchment areas were preliminarily calibrated using the weighted average method. Characteristic parameters of each catchment and Manning's coefficients were set according to the user manual. Then, observation data of the rainfall event on May 23, 2023, were used to calibrate the model. Figure 3 illustrates the comparison between the measured water level and simulated water level. The main adjustment was made to the Manning's coefficients of the channels, which were adjusted to values between 0.013 and 0.020. The Nash-Sutcliffe efficiency coefficients of the observed and simulated water levels at L1 and L2 were 0.79 and 0.89, with relative errors of 0.70% and 0.26% in the peak water depth. The error of peak appearance time was 15 minutes in both cases. The model verification results show that the stormwater flood model is suitable for subsequent research.





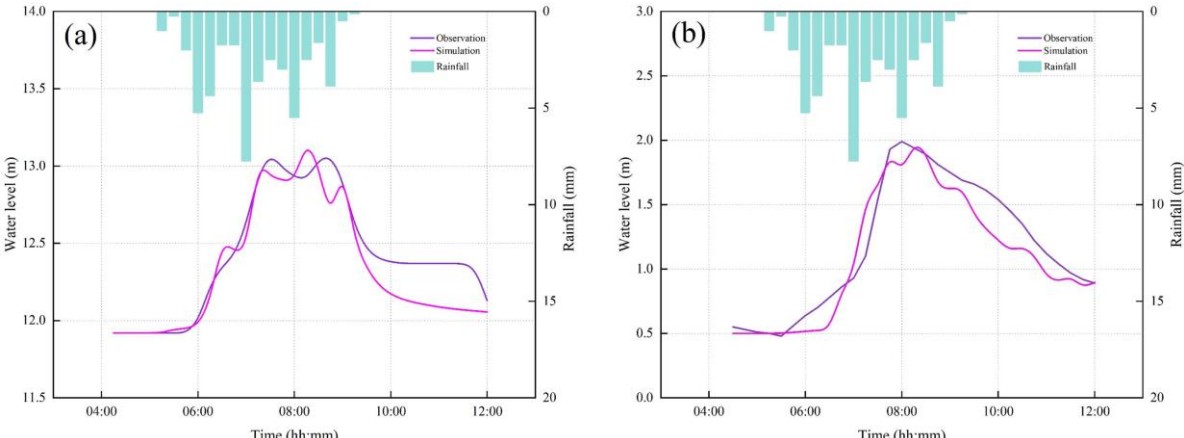

**Figure 3: Comparison of observed and simulated water levels at two monitoring points of the 20230523 rainstorm: (a) L1 for manhole level; (b) L2 for river level.**

### 3.3 Combination design of rainfall and tidal level

**3.3.1 Correlation analysis**

The correlation between two random variables, $X$ and $Y$, where $X=(X_1, X_2,\ldots, X_n)$ and $Y=(Y_1, Y_2,\ldots, Y_n)$, is assessed using Pearson correlation coefficient ($r$), Kendall rank correlation coefficient ($\tau$), and Spearman rank correlation coefficient ($\rho$). The calculation formulas are as follows:

$$r = \frac{\sum_{i=1}^{n}\left(X_i - \bar{X}\right)\left(Y_i - \bar{Y}\right)}{\sqrt{\sum_{i=1}^{n}\left(X_i - \bar{X}\right)^2}\sqrt{\sum_{i=1}^{n}\left(Y_i - \bar{Y}\right)^2}} \tag{4}$$

$$\tau = \frac{\sum_{i<j} sign\left[\left(X_i - X_j\right)\left(Y_i - Y_j\right)\right]}{n(n-1)/2} \tag{5}$$

$$sign\left[\left(X_i - X_j\right)\left(Y_i - Y_j\right)\right] = \begin{cases} 1 & X_i \leq X_j, Y_i \leq Y_j \text{ or } X_i \geq X_j, Y_i \geq Y_j \\ -1 & \text{other} \end{cases} \tag{6}$$

$$\rho = \frac{\sum_{i=1}^{n}\left(R_i - \bar{R}\right)\left(Q_i - \bar{Q}\right)}{\sqrt{\sum_{i=1}^{n}\left(R_i - \bar{R}\right)^2}\sqrt{\sum_{i=1}^{n}\left(Q_i - \bar{Q}\right)^2}} \tag{7}$$





where $R_i$ represents the rank of $X_i$ within $(X_1, X_2, \ldots, X_n)$, and $Q_i$ represents the rank of $Y_i$ within $(Y_1, Y_2, \ldots, Y_n)$.

The values of all the three indices range between [-1, 1]. If the index value is greater than 0, it indicates a positive rank
correlation between the two variables. If the index value is less than 0, it suggests a negative rank correlation. The larger the absolute value of the index, the stronger the rank correlation between the two variables.

**3.3.2 Copula selection**

The joint distributions between rainstorms and the tide levels are constructed using Copula functions. First, the Generalized Extreme Value (GEV) distribution, Normal (Norm) distribution, Gamma distribution, and Weibull distribution (Equations
(8)-(11)) are applied to estimate the marginal distributions for rainfall and tidal level separately. The best-fitting distribution functions are selected from these estimations. Then, two-dimensional Gaussian Copula, Student's t-Copula, Frank Copula, and Gumbel Copula (Equations (12)-(15)) are constructed to fit the best-fitting marginal distributions, generating the joint distributions for rainfall and tidal data. Based on this optimal joint distribution, the study examines how rainfall and tidal level vary under different conditions.

$$F(x) = \exp\left\{ -\left[ 1 - k\left( \frac{x - \mu}{\alpha} \right) \right]^{1/k} \right\} \tag{8}$$

$$F(x) = \frac{1}{\sqrt{2\pi}\sigma} \int_{-\infty}^{x} e^{-\frac{(t-\mu)^2}{2\sigma^2}} \, \mathrm{d}x \tag{9}$$

$$F(x) = \int_{0}^{x} \frac{x^{\beta-1}}{\alpha^{\beta}\Gamma(\beta)} e^{-\frac{x}{\alpha}} \mathrm{d}x \tag{10}$$

$$F(x) = 1 - e^{-\left( \frac{x-m}{a} \right)^b} \tag{11}$$

where $\mu$ and $m$ are location parameters; $a$ is the scale parameter; k, $\sigma$, $\beta$ and $b$ are shape parameters.

$$C(u,v) = \int_{-\infty}^{\Phi^{-1}(u)} \int_{-\infty}^{\Phi^{-1}(v)} \frac{1}{2\pi\sqrt{1-\rho^2}} \exp\left( \frac{-\left( r^2 + s^2 - 2\rho rs \right)}{2\left(1-\rho^2\right)} \right) \mathrm{d}r\mathrm{d}s \tag{12}$$

$$C(u,v,\rho,v) = \int_{-\infty}^{T_v^{-1}(u)} \int_{-\infty}^{T_v^{-1}(v)} \frac{1}{2\pi\sqrt{1-\rho^2}} \exp\left( 1 + \frac{r^2 + s^2 - 2\rho rs}{v(1-\rho)^2} \right)^{-\frac{v+2}{2}} \mathrm{d}r\mathrm{d}s \tag{13}$$





$$C(u,v) = -\frac{1}{\theta} \ln\left[ 1 + \frac{\left(e^{-\theta u} - 1\right)\left(e^{-\theta v} - 1\right)}{e^{-\theta} - 1} \right] \tag{14}$$

$$C(u,v) = \exp\left\{ -\left[ (-\ln u)^{\theta} + (-\ln v)^{\theta} \right]^{1/\theta} \right\} \tag{15}$$

where $u$ and $v$ represent the marginal distributions of variables; $\Phi^{-1}$ is the inverse function of the one-dimensional standard normal distribution; $\rho$ is the linear correlation coefficient; and $\theta$ is the parameter of the Copula function's generator.

Both marginal and joint distributions are estimated using the maximum likelihood method to estimate their parameters. Commonly used methods for selecting the best-fitting marginal distribution functions and Copula functions include the Root Mean Square Error (RMSE) method, the Akaike Information Criterion (AIC) method, and the Bayesian Information Criterion (BIC) method. The calculation formulas are as follows:

$$RMSE = \sqrt{\frac{1}{n} \sum_{i=1}^{n} \left( p_i - p_{ei} \right)^2} \tag{16}$$

where $p_i$ and $p_{ei}$ represent the theoretical and empirical frequencies of the joint distribution, respectively.

$$AIC = -2\ln(l_{\max}) + 2k \tag{17}$$

$$BIC = 2N\ln(l_{\max}) + k\ln N \tag{18}$$

where $N$ represents the sample size; $k$ represents the number of variables; and $l_{max}$ is the maximum likelihood function value. The three methods mentioned above can help identify functions with better fitting, but they do not necessarily provide statistical evidence for whether a particular function is suitable for describing the multivariate event. Here, the commonly used Kolmogorov-Smirnov (K-S) test method is introduced. The K-S test is a widely used non-parametric test method, and the test statistic $D$ is defined as follows:

$$D = \max_{1 \le k \le n} \left\{ \left| C_k - \frac{i}{n} \right|, \left| C_k - \frac{i-1}{n} \right| \right\} \tag{19}$$

where $C_k$ represents the theoretical distribution of the observed samples; $i$ is the index of the observed samples after being sorted in ascending order; $n$ is the sample size. When $D$ is less than the critical value $D_\alpha$, the test is accepted.





### 3.3.3 Joint risk probability analysis

In the case of multivariate hydrological events, where $X$ and $Y$ are random variables with marginal distribution functions $F_X(x)$ and $F_Y(y)$, and joint distribution function $F(x, y)$, and with specified thresholds $x^*$ and $y^*$, multivariate return periods are
typically defined in two types.

The joint risk rate calculates the probability that at least one of the variables ($X$ or $Y$) exceeds the specified threshold. The return period calculated based on this concept is referred to as the "Or" return period ($R_{Or}$), and the calculation formula is as follows:

$$R_{Or} = \frac{1}{P((X > x^*) \bigcup (Y > y^*))} = \frac{1}{1 - F(x^*, y^*)} \tag{20}$$

The joint occurrence probability calculates the probability that both $X$ and $Y$ exceed their respective specified thresholds. The return period calculated based on this concept is referred to as the "And" return period ($R_{And}$), and the calculation formula is as follows:

$$R_{And} = \frac{1}{P((X > x^*) \bigcap (Y > y^*))} = \frac{1}{1 - F_X(x^*) - F_Y(y^*) + F(x^*, y^*)} \tag{21}$$

In both of these multivariate return periods, it is possible for different variable combinations to have the same Copula
function value, indicating the same return period. This means that traditional return period methods cannot distinguish the differences between these combinations. In Fig. 4(a), the green area represents the danger zone for the "And" return period, where point A is located. It's evident that the joint probability at point B is higher than point A, which implies that point B is considered a dangerous event. However, in the "And" return period, the danger zone doesn't include point B. Therefore, the "And" return period narrows down the danger area. Conversely, analyzing Fig. 4(b), the yellow area corresponds to the
danger zone for the "Or" return period. It's noticeable that point B has one variable significantly large while the other one quite small. In engineering applications, this might not be considered as a significant risk. Nevertheless, in the "Or" return period, point B is included in the danger zone.

To address this issue, Salvadori and De Michele introduced the concept of Kendall return periods (Salvadori and De Michele, 2010)。They defined a boundary, represented by a curve $C(u,v)=p$, for events with the same Copula value. This boundary
divides the event domain into safe and dangerous areas. A larger $p$ value results in a smaller dangerous area, while dangerous areas with smaller $p$ values will necessarily cover those with larger $p$ values. As shown in Fig. 4 (c), the yellow dangerous area is included in the green range. Therefore, the bivariate return period under the condition $C(u,v)=p$ can be called "Kendall" return period ($R_{Kendall}$) and defined as follows:

$$R_{Kendall} = \frac{1}{P\left[C(u,v) > p\right]} = \frac{1}{1 - P\left[C(u,v) \le p\right]} = \frac{1}{1 - K_c(p)} \tag{22}$$

where $K_c(p)$ is the Kendall distribution function. For Gaussian Copula and Student's t-Copula functions, there is no explicit

formula to calculate $K_c(p)$. Therefore, it is often computed through Monte Carlo simulations, as suggested by Zhang et

al.(Zhang et al., 2022).

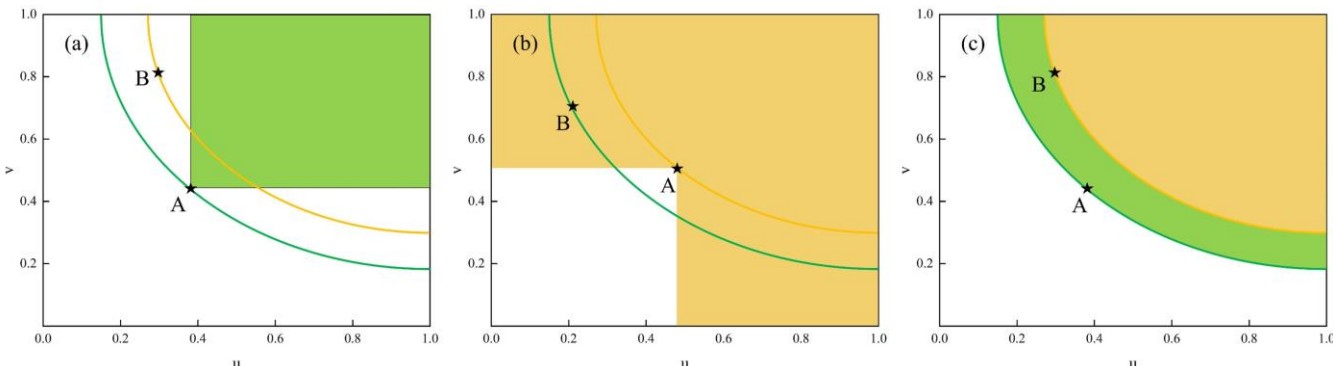

**Figure 4: The schematic diagram of (a) $R_{And}$; (b) $R_{Or}$; (c) $R_{Kendall}$.**


### 3.3.4 Designing of combined rainfall and tide events

In the case of the bivariate joint distribution, the design value corresponding to a specific return period is composed of

numerous variable combinations. Among these combinations, there is always one that maximizes the joint probability

density. This is referred to as the Maximum Possible Weighting Function (MPWF) method, and the calculation formula is as

follows:

$$\left(u_m, v_m\right) = \arg\max f(x, y) \tag{23}$$

$$f(x, y) = c\left(u_x, v_y\right) \cdot f(x) \cdot f(y) \tag{24}$$

where $c(u_x, u_y)$ represents the probability density function of Copula function; $f(x)$ and $f(y)$ represent the probability density

functions of rainfall and tide, respectively.

In this study, the maximum rainfall of different durations, including 1-hour (1-h), 3-hour (3-h), 6-hour (6-h), 12-hour (12-h)

and 24-hour (24-h) duration, are calculated year by year. Take the minimum value from the set of maximum values as the

threshold. Select rainfall samples that exceed this threshold $P$ and the highest tidal levels of the corresponding day $Z$ as the

variable. Determine the optimal marginal distributions and joint distribution. For a certain joint probability, find the specific




combination that maximizes the joint probability density. Once this combination is identified, the design values for rainfall
and tidal levels are computed using the inverse functions of the optimal marginal distributions.

These design values are determined for different durations above and different return periods, including 2-year (2-yr), 3-year
(3-yr), 5-year (5-yr), 10-year (10-yr), 20-year (20-yr), 50-year (50-yr), 100-year (100-yr) and 200-year (200-yr) return period
(RP). For short durations (1 h and 3 h), the Pilgrim & Cordery rainfall model is used to construct the rainfall processes. For
longer durations (6 h, 12 h, and 24 h), the design rainfall patterns are derived using the same-frequency method, referencing
the outcomes of the "Technical Report on the Compilation of Guangzhou Rainstorm Intensity Formula and Design Rainfall
Patterns" for calculations.

The design tidal level processes are typically created using the equal-multiple method. However, when the typical tidal
processes contain negative value, the magnification factor tends to lower the low tide levels, which leads to an exaggerated
tidal range. In this study, a modified equal-multiple method is employed to design the tidal level process. Firstly, a typical
tidal event is selected based on an extensive dataset of tidal observations. Then, the design tidal process is calculated
according to the following formula while controlling for the high tidal levels:

$$
\begin{cases}
Z_i \geqslant \overline{Z}, & Z_{p,i} = \left( Z_i - \overline{Z} \right) \times k_{p,1} + \overline{Z} \\
Z_i < \overline{Z}, & Z_{p,i} = Z_i \times k_{p,2} + Z_i
\end{cases}
\tag{25}
$$

$$
k_{p,1} = \frac{Z_p}{Z_{\max}}
\tag{26}
$$

$$
k_{p,2} = \frac{Z_{\min}}{Z_p}
\tag{27}
$$

where $Z_i$ represents the typical tidal level at time $i$; $\overline{Z}$ is the average tidal level of the typical tidal process; $Z_{p,\,i}$ is the
designed tidal level at time $i$ for the $p$ frequency; $k_{p,\,1}$ is the ratio of the designed high tide level to the typical tidal level for
the $p$ frequency; $k_{p,\,2}$ is the ratio of the typical tidal level to the designed tidal level for the $p$ frequency; $Z_{max}$ is the maximum
tidal level value in the typical tidal pattern; $Z_{min}$ is the minimum tidal level value in the typical tidal pattern; $Z_p$ represents the
designed tidal level.

The above method may result in differences between the maximum value and the design values. Therefore, further
adjustments are made to the designed tide process to obtain the final design tidal level process.

$$
k = \frac{Z_p}{\max \left( Z_{p,i} \right)}
\tag{28}
$$



$$Z'_{p,i} = k \times Z_{p,i} \tag{29}$$

where $k$ is the correction coefficient, and $Z'_{p,i}$ is the final designed process.

### 3.4 Quantify the impact of rainfall and tide on flooding

Taking the severity of flooding as the evaluation index, the influence of rainfall and tide on flooding is estimated. Flooding severity is represented by the flooding volume. Through the storm flood model, variations in flooding volume concerning rainfall and tidal levels are obtained. There is still one thing should be concerned. Taking into account the lag effect of runoff processes, the simulation time is extended by an additional 3 hours on top of the rainfall duration in scenario simulation. To quantify the impact of rainfall and tidal levels on flooding, this study introduces the impact degree index.

Firstly, calculate the range of rainfall and tidal level design values for different durations from 2-yr RP to 200-yr RP using the following formula:

$$\Delta x_t = \frac{X_{2,t} - X_{1,t}}{X_{2,t}} \tag{30}$$

$$\Delta y_t = \frac{Y_{2,t} - Y_{1,t}}{Y_{2,t}} \tag{31}$$

where $X_{1,t}$ and $X_{2,t}$ represent the design values of rainfall for durations of $t$ with 2-yr RP and 200-yr RP, respectively; $Y_{1,t}$ and $Y_{2,t}$ represent the design values of tidal levels for durations of $t$ with 2-yr RP and 200-yr RP, respectively.

Next, calculate the relative changes in flooding volume when tidal levels remain 2-yr RP and 200-yr RP respectively, rainfall varies from 2-yr RP to 200-yr RP. Then calculate the average of these relative changes. Similarly, calculate the relative change in flooding volume when rainfall remain 2-yr RP and 200-yr RP respectively, tidal levels vary from 2-yr RP to 200-yr RP. Calculate the average of these relative changes as well.

$$\Delta Vx_{1,t} = \frac{|Vx_{21,t} - Vx_{11,t}|}{\max(Vx_{21,t}, Vx_{11,t})} \tag{32}$$

$$\Delta Vx_{2,t} = \frac{|Vx_{22,t} - Vx_{12,t}|}{\max(Vx_{22,t}, Vx_{12,t})} \tag{33}$$

$$\Delta Vx_t = \frac{\Delta Vx_{2,t} + \Delta Vx_{1,t}}{2} \tag{34}$$





$$\Delta V y_{1,t} = \frac{\left| V x_{12,t} - V x_{11,t} \right|}{\max\left( V x_{12,t}, V x_{11,t} \right)} \tag{35}$$

$$\Delta V y_{2,t} = \frac{\left| V x_{22,t} - V x_{21,t} \right|}{\max\left( V x_{22,t}, V x_{21,t} \right)} \tag{36}$$

$$\Delta V y_{t} = \frac{\Delta V y_{2,t} + \Delta V y_{1,t}}{2} \tag{37}$$

where $Vx_{11,t}$ and $Vx_{12,t}$ represent the simulated flooding volume for a duration of $t$ when rainfall is at 2-yr RP and tidal levels are at both 2-yr RP and 200-yr RP; $Vx_{21,t}$ and $Vx_{22,t}$ represent the simulated flooding volume for a duration of $t$ when rainfall is at 200-yr RP and tidal levels are at both 2-yr RP and 200-yr RP; $\Delta Vx_t$ is the variation in flooding volume due to rainfall, and $\Delta Vy_t$ is the variation in flooding volume due to tidal levels.

Finally, the impact factors $Dx_t$ and $Dy_t$ of rainfall and tidal levels on flooding volume are computed. These impact factors are then normalized to obtain the rainfall's impact degree index on regional flooding, which is represented as $D_t$. The calculation formulas are as follows:

$$Dx_{t} = \frac{\Delta V x_{t}}{\Delta x_{t}} \tag{38}$$

$$Dy_{t} = \frac{\Delta V y_{t}}{\Delta y_{t}} \tag{39}$$

$$D_{t} = \frac{Dx_{t}}{Dx_{t} + Dy_{t}} \tag{40}$$

where $Dx_t$ and $Dy_t$ represent the variation in flooding volume caused by unit variation in rainfall and tidal levels, respectively. This study considers only the influence of rainfall and tidal levels as natural factors. Therefore, the impact index of tidal levels on flooding is calculated as $1 - D_t$. The range of values for $D_t$ is 0 to 1, where 0 indicates being solely influenced by tidal levels, and 1 indicates being solely influenced by rainfall. A higher value of $D_t$ signifies a stronger influence of rainfall and a weaker influence of tidal levels.

### 3.5 Spatial interaction of drainage units

Considering the topography of the watershed and in combination with the direction of the drainage network, the drainage units are divided. The outlet number data from urban drainage network data are recorded in the outlet number set. The first inspection well number $Y_i$ is searched upstream of the outlet number $O_i$, and then the pipelines with $Y_i$ as the downstream





inspection well are searched. The search results are checked for null values. If they are null, the search for the outlet $O_i$ is completed, and the process proceeds to the next outlet. All the inspection wells, pipeline segments, and sub-watersheds upstream of the outlet form a drainage unit. The main drainage channels of the drainage network along the river and their inspection wells are assigned to various adjacent drainage units. In this study, a total of 22 drainage units are defined, as

shown in Fig. 5.

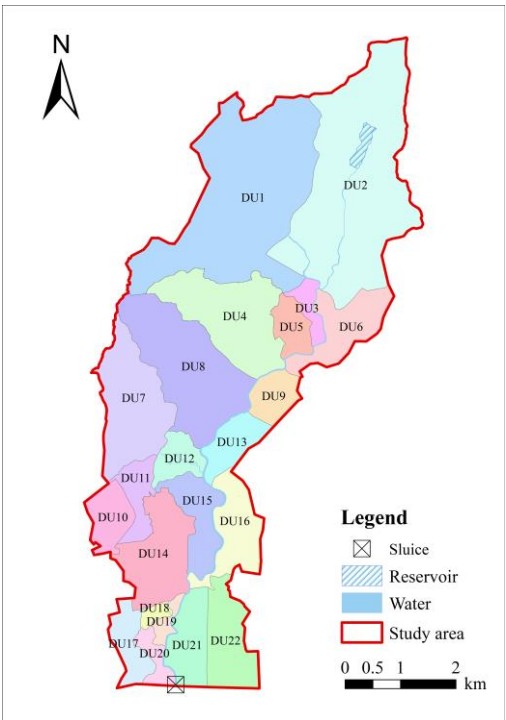

**Figure 5: Diagram of drainage units in study area.**

The urban gravity model is developed based on Newton's universal law of gravitation from physics and has become an

important tool in quantitative research in the fields of economic geography, urban planning and regional analysis. In this study, the gravity model is used to explore the spatial interaction forces among the 22 drainage units. In the gravity model approach, the interaction force between two drainage units is positively correlated with their mass and inversely proportional to the square of the distance. The formula for calculation is as follows:

$$T_{ij} = k \frac{Q_i^{\alpha} Q_j^{\beta}}{d_{ij}^2} \tag{41}$$





where $T_{ij}$ represents the spatial interaction value between drainage unit $i$ and drainage unit $j$; $d_{ij}$ represents a comprehensive distance index from the center of drainage unit $i$ to the center of drainage unit $j$; $Q_i$ and $Q_j$ represent the masses of drainage unit $i$ and drainage unit $j$, respectively; $k$, $\alpha$ and $\beta$ are coefficients and set to 1.

The length of the drainage network represents the drainage capacity, while the flooding volume indicates the degree of drainage obstruction. In this study, the percentage of the drainage network length of the drainage unit and flooding volume are combined to serve as the quality indicator in the urban gravity model. The formula for the calculation is as follows:

$$Q_i = \frac{l_i}{lV_i} \tag{42}$$

where $l_i$ represents the length of the drainage network for drainage unit $i$; $l$ represents the total length of the drainage network in the study area; $V_i$ represents the flooding volume of drainage unit $i$.

### 3.6 The interaction between fluvial flooding and pluvial flooding

The south side of the study area is adjacent to the Pearl River. During rainfall, the Shahe River carries a significant overall flow, typically accompanied by downstream tidal surges. In a short period, the river water level rapidly rises. Most of the stormwater outlets have lower elevations, which means they get inundated by the river water. This severely hampers drainage and even leads to reverse flow, exacerbating urban flooding. The flood events combined by fluvial flooding and pluvial flooding are divided into five stages, with key time points being the river water level exceeding the outlet elevation and riverbank overtopping. Stage 1(FP_S1) means that the river water level is below the outlet elevation. Stage 2(FP_S2) means that the river water level exceeds the elevation of the drainage outlet until the river overflows its bank. Stage 3(FP_S3) is the riverbank overtopping stage. Stage 4(FP_S4) means that the riverbank overtopping continues until the river water level drops below the outlet elevation. Stage 5(FP_S5) means that the river water level recedes and does not exceed the outlet elevation again until the simulation ends. In cases where the outlets are not connected to the river, FP_S2 to FP_S4 would not exist. This study introduces the scenarios in which the outlets are not connected to the river for comparative analysis of the interactions between fluvial flooding and pluvial flooding.

Firstly, the following formula is used to calculate the flooding volume $V_t$ when the outlets are connected with the river:

$$V_t = FP\_S1_t + FP\_S2_t + FP\_S3_t + FP\_S4_t + FP\_S5_t \tag{43}$$

where $FP\_S1_t$, $FP\_S2_t$, $FP\_S3_t$, $FP\_S4_t$ and $FP\_S5_t$ respectively represent the flooding volume with a duration of $t$ in each stage.

The percentage of fluvial flooding volume to total flooding volume is expressed by $F\_S3_t$, and the calculation formula is as follows:





$$F\_S3_t = \frac{V_{F\_S3_t}}{V_t} \qquad (44)$$

where $V_{F\_S3_t}$ represent the fluvial flooding volume with a duration of $t$ in FP_S3.

Then calculate the relative change $FP_t$ of flooding volume whether the outlets are connected with the river, and the calculation formula is as follows:

$$FP_t = \frac{(PF_t - V_t)}{V_t} \qquad (45)$$

where $PF_t$ represents the flooding volume which lasts for $t$ hours and the outlets are not connected with the river.

The simulation that the outlets are not connected with the river will not produce fluvial flooding, only pluvial flooding.

Therefore, the difference between $FP_t$ and $F\_S3_t$ represents the percentage of flooding volume due to outlets being submerged compared to the total volume.

## 4 Results and analysis

### 4.1 Joint distribution and combination design of rainfall and tide

Statistically analyze the daily rainfall and the corresponding maximum tidal level data, and calculate the $r$, $\tau$ and $\rho$

respectively, and the results are 0.89, 0.83 and 0.88. All the three indicators are close to 1, indicating that there is a significant positive correlation.

The *RMSE*, *AIC*, *BIC*, and *K-S* for different marginal distributions of individual variables are calculated and presented in Table 1. From Table 1, it can be observed that for the rainfall, GEV distribution has the smallest *RMSE*, lowest *AIC*, lowest *BIC*, and passes the *K-S* test. This indicates that the GEV distribution is the optimal marginal distribution function for this

variable. For the 1-h, 3-h, and 24-h durations, the optimal marginal distributions for the tidal levels are the Gamma distribution, while for the 6-h and 12-h durations, they are the Weibull distributions. Figure 6 displays the fitting diagram of the optimal marginal distribution and empirical distribution for the 24-h rainfall and tidal level, demonstrating a good fit of the theoretical distribution functions.





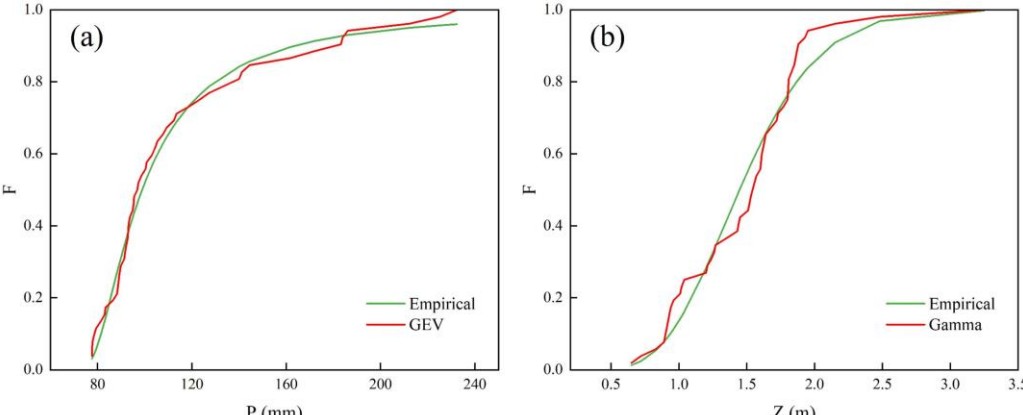

**Figure 6: The optimal marginal distribution of (a) P; (d) Z.**

**Table 1: The goodness-of-fit test for the marginal distribution functions**

| Times(h) | Marginal distributions | P | | | | Z | | | |
|---|---|---|---|---|---|---|---|---|---|
| | | RMSE | AIC | BIC | K-S | RMSE | AIC | BIC | K-S |
| 1 | GEV | **0.035** | **335.690** | **341.303** | **0.104** | 0.026 | 49.299 | 54.912 | 0.083 |
| | Norm | 0.093 | 357.198 | 360.940 | 0.188 | 0.590 | 49.716 | 53.458 | 1.000 |
| | Gamma | 0.082 | 350.256 | 353.998 | 0.188 | **0.026** | **47.079** | **50.821** | **0.083** |
| | Weibull | 0.099 | 364.278 | 368.020 | 0.188 | 0.035 | 50.007 | 53.750 | 0.083 |
| 3 | GEV | **0.037** | **325.464** | **330.454** | **0.103** | 0.044 | 41.501 | 46.492 | 0.128 |
| | Norm | 0.082 | 345.930 | 349.257 | 0.154 | 0.591 | 42.596 | 45.923 | 1.000 |
| | Gamma | 0.066 | 336.618 | 339.945 | 0.128 | **0.045** | **39.488** | **42.815** | **0.128** |
| | Weibull | 0.090 | 351.458 | 354.786 | 0.205 | 0.048 | 42.871 | 46.198 | 0.128 |
| 6 | GEV | **0.031** | **437.009** | **442.685** | **0.122** | 0.050 | 56.114 | 61.790 | 0.122 |
| | Norm | 0.086 | 460.232 | 464.016 | 0.163 | 0.588 | 54.676 | 58.460 | 1.000 |
| | Gamma | 0.066 | 449.016 | 452.799 | 0.122 | 0.054 | 54.531 | 58.314 | 0.102 |
| | Weibull | 0.086 | 462.671 | 466.455 | 0.163 | **0.043** | **54.200** | **57.984** | **0.122** |
| 12 | GEV | **0.025** | **449.977** | **455.653** | **0.082** | 0.061 | 64.034 | 69.709 | 0.143 |
| | Norm | 0.111 | 485.466 | 489.249 | 0.224 | 0.587 | 62.417 | 66.200 | 1.000 |
| | Gamma | 0.090 | 471.221 | 475.005 | 0.184 | 0.069 | 62.773 | 66.557 | 0.122 |
| | Weibull | 0.108 | 485.460 | 489.244 | 0.224 | **0.054** | **61.608** | **65.392** | **0.143** |
| 24 | GEV | **0.031** | **491.344** | **497.198** | **0.077** | 0.059 | 71.614 | 77.468 | 0.154 |
| | Norm | 0.124 | 533.271 | 537.173 | 0.231 | 0.588 | 72.491 | 76.393 | 1.000 |
| | Gamma | 0.106 | 517.973 | 521.876 | 0.212 | **0.058** | **69.048** | **72.951** | **0.154** |
| | Weibull | 0.118 | 532.585 | 536.487 | 0.212 | 0.054 | 73.716 | 77.619 | 0.173 |





Upon using the optimal marginal distribution functions, two-dimensional joint probability distribution models for rainfall
and tidal level are constructed using Copula functions. The *RMSE*, *AIC*, *BIC*, and *K-S* for each distribution are presented in
Table 2. All the distributions passed the *K-S* test. When the duration is 1 hour, the Gumbel Copula function has the lowest
*AIC* and *RMSE*. For 3-h and 24-h durations, the Frank Copula function is the optimal Copula distribution. For 6-h and 12-h
durations, the Gaussian Copula provides the best joint probability distribution model for rainfall and tidal level. Figure 7(a)
displays the fitting diagram of the optimal Copula distribution function and the empirical distribution function for 24-h
duration, with an $R^2$ of 0.97, indicating a good fit. Figure 7(b) shows the two-dimensional joint probability distribution of
24-h duration, providing insight into the joint distribution probabilities of different combinations.

**Table 2 The goodness-of-fit test for the joint distribution functions of P and Z**

| Times(h) | Copula | RMSE | AIC | BIC | K-S |
|---|---|---|---|---|---|
| 1 | Gaussian | 0.035 | -0.041 | 1.830 | 0.125 |
| | t | 0.035 | 3.956 | 9.570 | 0.125 |
| | Frank | 0.035 | -0.161 | 1.710 | 0.125 |
| | **Gumbel** | **0.036** | **-1.004** | **0.867** | **0.125** |
| 3 | Gaussian | 0.041 | 0.711 | 2.663 | 0.154 |
| | t | 0.041 | 1.751 | 7.605 | 0.154 |
| | **Frank** | **0.041** | **0.312** | **2.264** | **0.154** |
| | Gumbel | 0.044 | 0.860 | 2.811 | 0.154 |
| 6 | **Gaussian** | **0.036** | **-0.793** | **1.099** | **0.122** |
| | t | 0.035 | 3.207 | 8.883 | 0.122 |
| | Frank | 0.038 | 0.352 | 2.244 | 0.143 |
| | Gumbel | 0.038 | -0.080 | 1.812 | 0.163 |
| 12 | **Gaussian** | **0.044** | **1.225** | **3.117** | **0.143** |
| | t | 0.045 | 4.736 | 10.411 | 0.163 |
| | Frank | 0.045 | 1.643 | 3.535 | 0.163 |
| | Gumbel | 0.045 | 1.691 | 3.582 | 0.184 |
| 24 | Gaussian | 0.043 | 0.711 | 2.663 | 0.154 |
| | t | 0.043 | 1.751 | 7.605 | 0.173 |
| | **Frank** | **0.043** | **0.312** | **2.264** | **0.154** |
| | Gumbel | 0.044 | 0.860 | 2.811 | 0.154 |





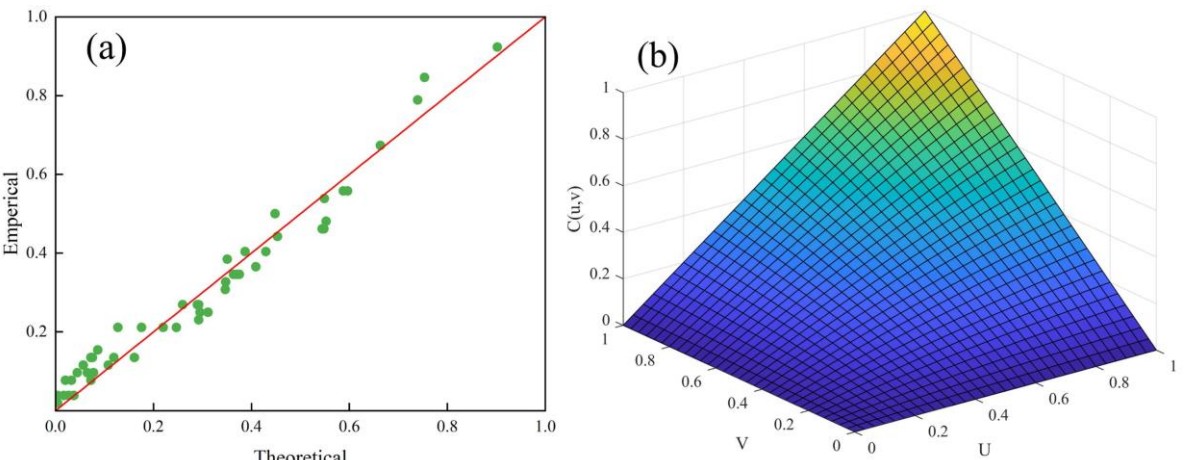

**Figure 7: The joint distribution results: (a) fitting diagram of the optimal Copula distribution function and the empirical distribution function; (b) joint probability distribution.**

Table 3 provides results for different types of bivariate return periods for 24-h duration, with rainfall and tidal level combinations that share the same univariate return period. From Table 3, it can be observed that the $R_{Or}$ for the rainfall-tide combination is smaller than the $R_{Kendall}$, and the $R_{Kendall}$ is smaller than the $R_{And}$. For instance, when the individual rainfall and tide return periods are both 10 years, the $R_{Kendall}$ for the rainfall-tide combination is 36.35 years, lying between the $R_{Or}$ (5.40) and the $R_{And}$ (66.88). The $R_{Kendall}$ defines dangerous areas based on joint probability values and provides a reasonable description of the return periods for various variable combinations. Based on the $R_{Kendall}$, Table 4 provides design values of rainfall and tide calculated using the MPWF method.

**Table 3 Comparison of different types of return period**

| RP(yr) | P(mm) | Z(m) | $R_{Or}$(yr) | $R_{And}$(yr) | $R_{Kendall}$(yr) |
|--------|-------|------|--------------|---------------|-------------------|
| 2 | 98.90 | 1.45 | 1.40 | 3.52 | 2.42 |
| 3 | 111.20 | 1.65 | 1.89 | 7.21 | 4.55 |
| 5 | 129.84 | 1.87 | 2.90 | 18.22 | 10.62 |
| 10 | 163.66 | 2.12 | 5.40 | 66.88 | 36.35 |
| 20 | 211.95 | 2.34 | 10.41 | 254.82 | 133.11 |
| 50 | 310.17 | 2.60 | 25.41 | 1543.63 | 785.87 |
| 100 | 424.12 | 2.79 | 50.41 | 6108.23 | 3082.09 |
| 200 | 589.74 | 2.97 | 100.41 | 24299.60 | 12205.61 |

**Table 4 Rainfall-tidal level design value**

| $R_{Kendall}$(yr) | P(mm) | | | | | Z(m) | | | | |
|-------------------|-------|-----|-----|------|------|------|-----|-----|------|------|
| | 1 h | 3 h | 6 h | 12 h | 24 h | 1 h | 3 h | 6 h | 12 h | 24 h |





| | | | | | | | | | |
|---|---|---|---|---|---|---|---|---|---|
| 2 | 41.93 | 66.19 | 70.85 | 77.71 | 91.93 | 1.20 | 1.23 | 1.38 | 1.50 | 1.47 |
| 3 | 44.02 | 70.55 | 76.40 | 83.23 | 98.40 | 1.33 | 1.36 | 1.51 | 1.63 | 1.64 |
| 5 | 46.81 | 75.66 | 82.98 | 90.13 | 107.64 | 1.46 | 1.51 | 1.64 | 1.75 | 1.78 |
| 10 | 50.74 | 83.36 | 93.28 | 101.21 | 121.39 | 1.63 | 1.65 | 1.75 | 1.87 | 1.95 |
| 20 | 55.94 | 91.76 | 103.67 | 113.89 | 136.15 | 1.77 | 1.77 | 1.86 | 1.96 | 2.10 |
| 50 | 65.50 | 101.98 | 119.80 | 131.33 | 164.72 | 1.95 | 1.93 | 1.97 | 2.10 | 2.23 |
| 100 | 77.12 | 111.79 | 132.42 | 149.13 | 182.37 | 2.06 | 2.03 | 2.05 | 2.17 | 2.39 |
| 200 | 84.19 | 122.51 | 147.74 | 170.31 | 211.79 | 2.33 | 2.12 | 2.11 | 2.24 | 2.49 |

The 1-h and 3-h design rainfall processes are determined using the Pilgrim & Cordery method, while the 6-h, 12-h, and 24-h
design rainfall patterns are obtained using the same frequency method. The tidal level process from May 22, 2020, was
chosen as the representative tidal level process. Figure 8 illustrates the design processes for various durations under the 200-
yr RP.

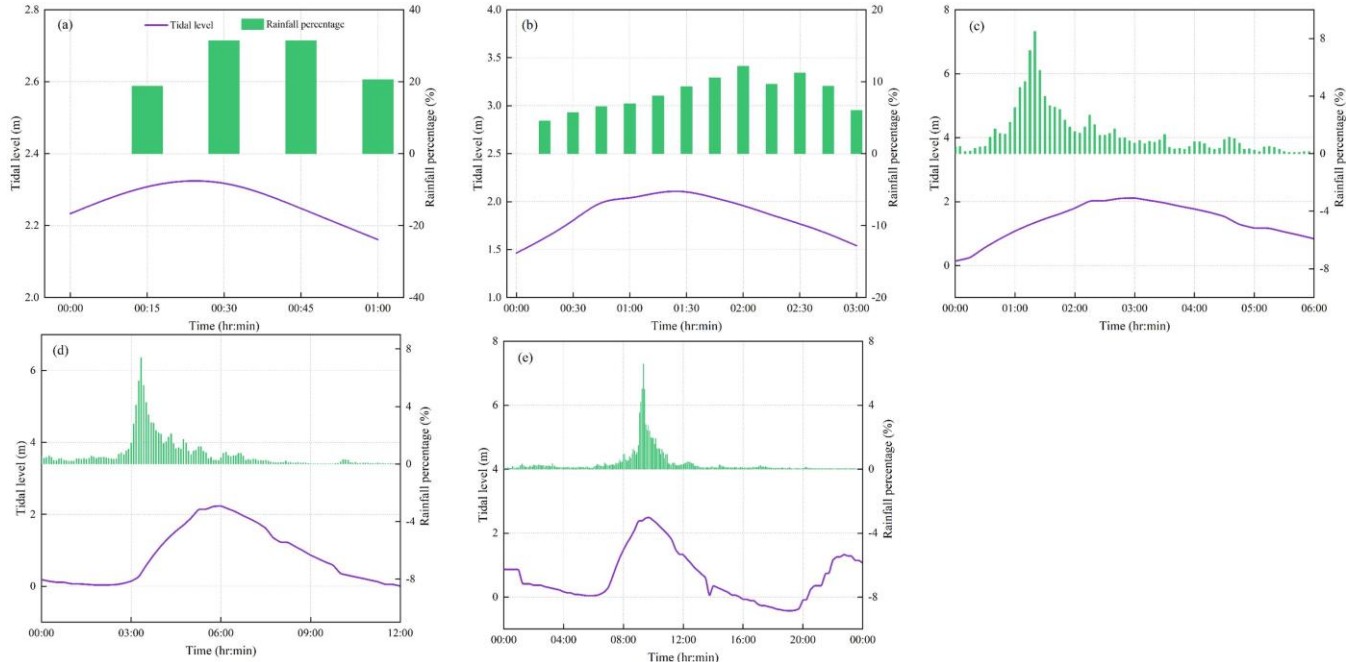

**Figure 8: Design process of P and Z with 200-yr RP: the sequence of duration from (a) to (e) are 1-h, 3-h, 6-h, 12-h and 24-h
durations.**

### 4.2 Analysis of the influence of rainfall and tide on flood

By using the stormwater flooding model, simulations were conducted for the specified rainfall and tide scenarios. The
comparison of flooding volumes for different scenarios is shown in Fig. 9. From Fig. 9, in general, the flooding volume
significantly increases with the increase in return period and duration. For the same return period, the flood volume caused




by the rainfall and tide with a longer duration is greater than those with a shorter duration. However, there are cases where shorter-duration scenarios result in larger flooding volumes than longer-duration scenarios, such as 6-h duration with a 20-yr RP compared to 12-h duration with a 20-yr RP. Similarly, for the same duration, scenarios with smaller return periods may yield larger flooding volumes than scenarios with larger return periods, such as 3-h duration with a 10-yr RP compared to a

20-yr RP. This indicates that the impact of rainfall and tidal levels on flood is influenced not only by the total volume but also by factors like the temporal distribution of rainfall and tide, and other factors.

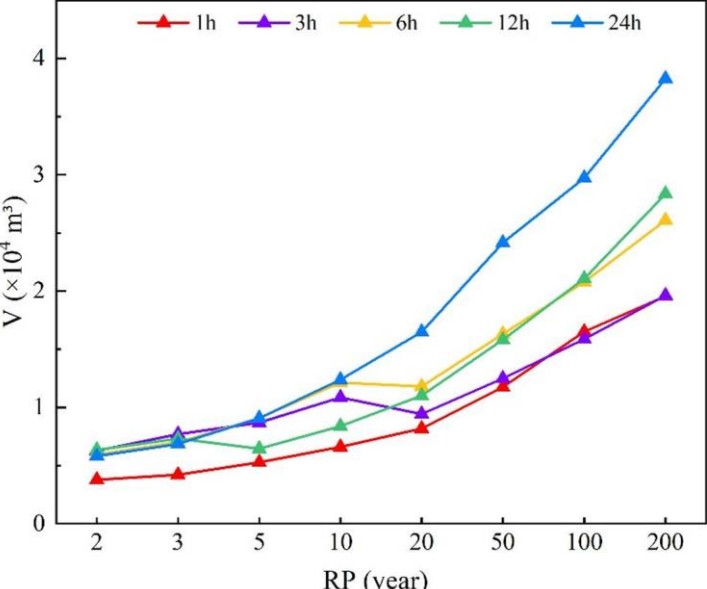

**Figure 9: Comparison of flooding volume.**

By analyzing the results of the maximum inundation depth, the spread of flooding in different areas can be assessed, and the regions potentially affected by flooding can be identified. The maximum inundation depth results for 2-yr RP of 1-h duration and 200-yr RP of 24-h duration scenarios are shown in the Fig. 10. As duration and return period increase, the inundation areas for DU1, DU2, DU4, DU5, DU10, and DU13 expand significantly, transitioning from moderate (purple waterlogging) to severe (red waterlogging) inundation levels. Severe flooding is observed in DU8, DU14 to DU22, while DU4 experiences

moderate flooding. DU21 and DU22 are the most severely affected by flooding.





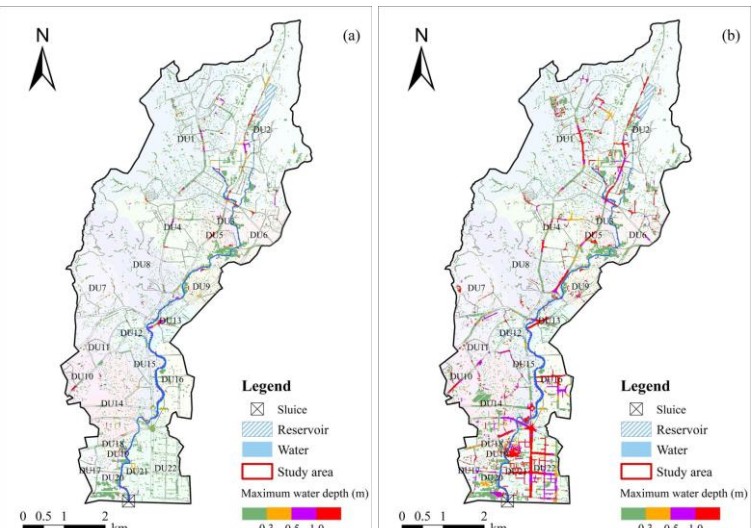

**Figure 10: Inundation extent and water depth under 24 h scenarios: (a) 2-yr RP; (b) 200-yr RP.**

Using the gravity model, spatial interaction forces between different drainage units were calculated. Taking the 200-yr RP

for 24-h duration as an example, the interaction forces between the drainage units on the left and right banks of the Shahe River are shown in Fig. 11. From Fig. 11, it is apparent that DU12 and DU9 have relatively strong interaction forces with other drainage units. This suggests that the drainage unit located in the middle region interacts significantly with the upstream and downstream units. DU12 has the strongest interaction force with the upstream DU7, followed by DU8. This implies that water from multiple drainage units flows into a single unit. In the downstream area, it has the strongest

interaction forces with DU14 and DU15, indicating that water is distributed from one unit to multiple units. There is also a relatively strong interaction force between the adjacent drainage units on the downstream left bank of the river. The results show that water levels between drainage units interact during flood events, affecting the extent and severity of flooding.





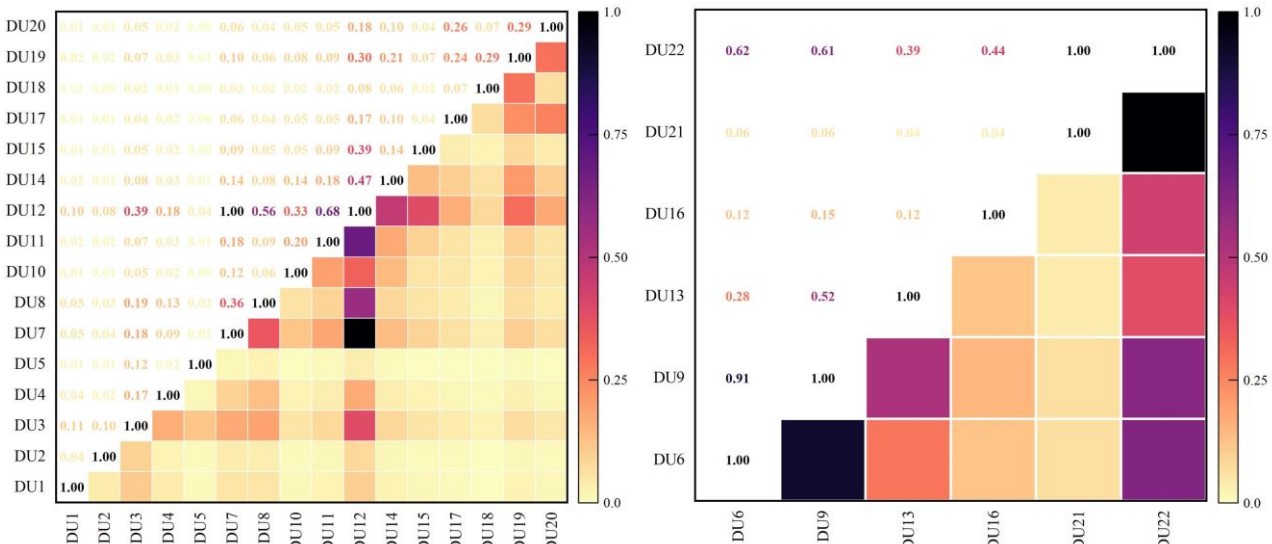

**Figure 11: Spatial interaction forces between different drainage units.**


Rainstorm and storm surge are natural factors contributing to flooding. This study quantifies the impact of rainfall and tide on flooding by $D_t$. The results of $D_t$ are presented in Fig. 12. DU1 to DU12 in Fig. 12(a) and DU13, DU14 and DU17 in Fig. 12(b) have $D_t$ values close to 1, indicating that drainage units far from the Pearl River are hardly affected by tides. For DU14 to DU16 and DU18 to DU22 in Fig. 12(b), $D_t$ values are around 0.5, which indicates that flooding is the result of the

combined effects of rainfall and tides. In spatial terms, the $D_t$ values for DU14 to DU16 and DU18 are greater than those for DU19 to DU22, indicating that as the distance between the drainage units and the Shahe River and Pearl River increases, the impact of tides diminishes. DU21 is most significantly affected by tides, as it is directly influenced by the tide at the drainage outlet, posing a risk of tidal backflow. Moreover, it is located adjacent to the river, making it indirectly affected by the tide. There's a general trend that longer duration results in smaller $D_t$ values compared to shorter durations, indicating that the

influence of tides on flooding is continually strengthening. Specifically, for DU20 and DU21, as the duration increases, $D_t$ changes from greater than 0.5 to less than 0.5. This shift suggests that the influence of tide becomes more significant than rainfall with longer durations. For all other drainage units, the impact of rainfall remains more significant at any duration.

Under short-duration, except for DU21, other drainage units show that the value of $D_t$ for 3-h duration is greater than that for 1-h duration, indicating an increase in the influence of rainfall. This suggests that the 1-h and 3-h tidal processes have

relatively similar impacts on flooding. However, despite the greater total rainfall amount in 3-h duration compared to 1-h duration, the higher average rainfall intensity could lead to more water entering the drainage system or river, resulting in more severe flooding issues. DU21 is notably influenced directly by tide. Under prolonged rainfall conditions, for DU19 to DU22, the influence of tide is minimal at 12-h duration and maximum at 24-h duration. These drainage units are located near the Pearl River estuary and are directly affected by tide. Comparing the 12-h and 6-h durations, the difference of tide impact



is small, while the impact of total rainfall amount is more significant. This is similar to the comparison between 3-h and 1-h durations. DU15 and DU18 have the lowest $D_t$ values at 12-h duration. Shorter-duration rainfall events often exhibit burst characteristics, potentially generating a significant amount of runoff in a short period, leading to intense and rapid rainfall-induced flooding. In contrast, longer-duration rainfall events may have reduced rainfall intensity but a longer duration, resulting in a greater total rainfall amount, which can still trigger flooding. The distribution of rainfall over a 12-h duration is

relatively even, and the rainfall intensity is not as intense as that in the case of 6-h heavy rainfall. Additionally, the total rainfall amount is less than that of a 24-h rainfall event. As a result, the impact of rainfall on flooding is relatively lower compared to the more intense 6-h and longer-duration 24-h rainfall events. Moreover, the 12-h duration provides more time for the tide rise to affect the coastal areas. When considering the entire region, it becomes evident that rainfall plays a significant role in all scenarios and the influence of tidal effects becomes most pronounced at 24-h duration.

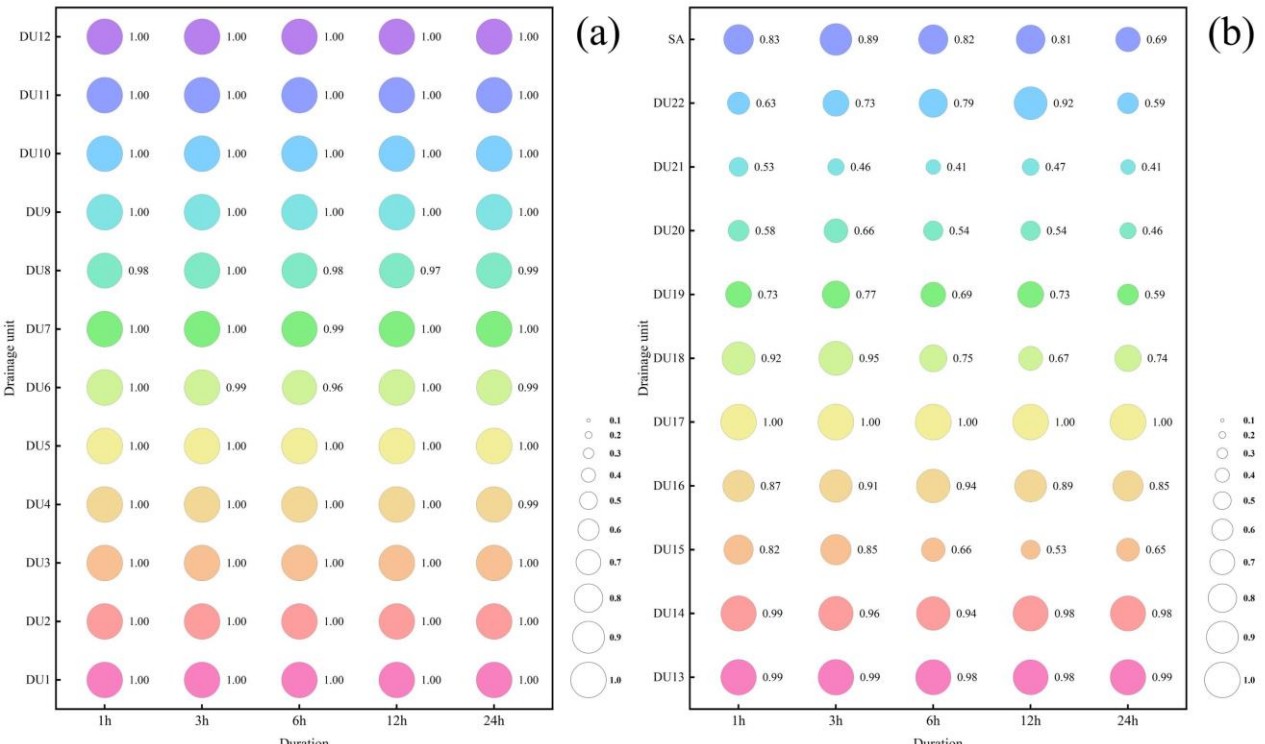


**Figure 12: Comparison of the impact degree indices.**

**4.3 Analysis of the interaction between fluvial flooding and pluvial flooding**

The time when the river water level exceeded the outfall elevation was recorded and summarized, as shown in Fig. 13. In the

1-h scenario with 2-yr RP, the outfall elevation is higher than the river water level, while in other scenarios, there is evidence of water topping over the outfalls. Generally, the average duration of water topping the outlets increases with longer durations and higher return periods. The large spacing between the upper and lower edges of the box plots, indicates



significant variability in the time during which each drainage outlet experiences being submerged by water. The outlet O1
was affected by river water level for 0.2 hours in the 200-yr RP of 24-h duration, while it was not impacted in other
scenarios. Conversely, outfall O2 was affected by river water for almost the entire simulation time.

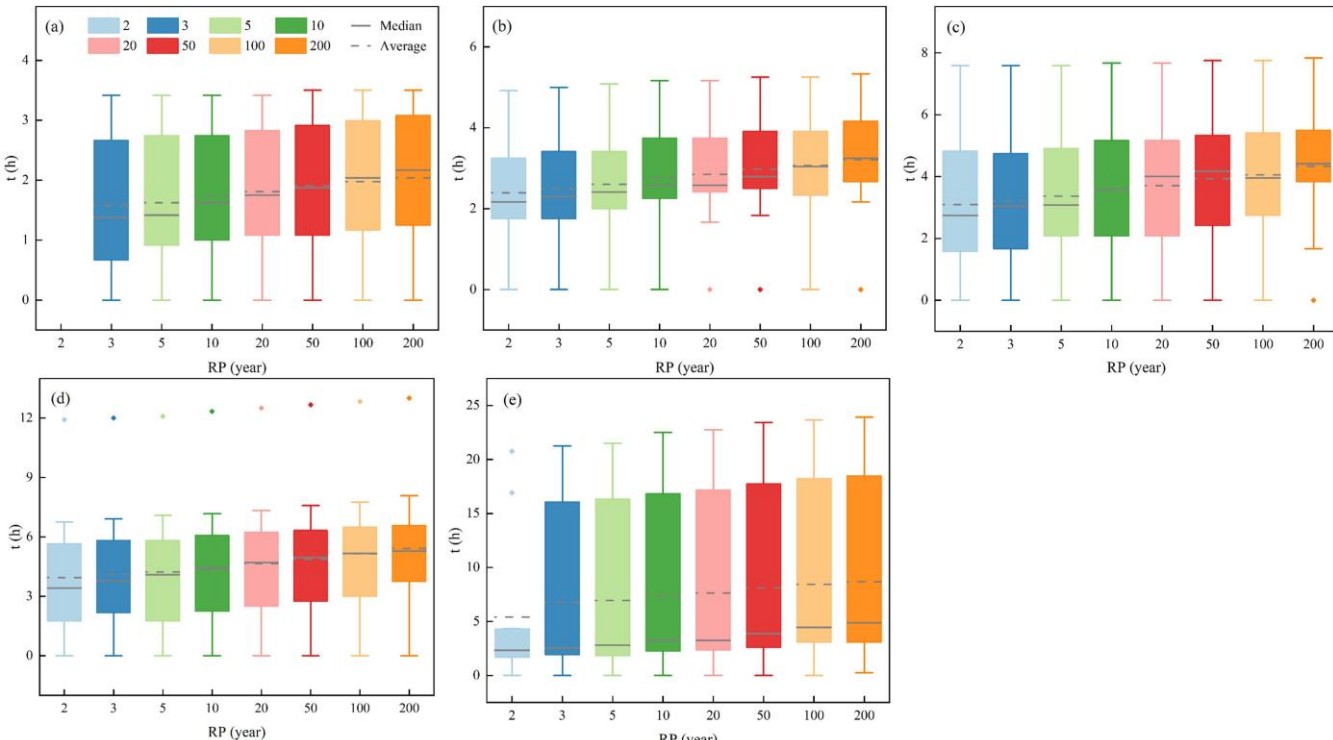

**Figure 13: The average time of water topping the outlets: the sequence of duration from (a) to (e) is 1h, 3h, 6h, 12h and 24h.**

Fluvial flooding may result from rainfall runoff and urban drainage systems. When the drainage system is efficient, it can
increase flood pressure downstream in the river, potentially leading to backflow, which may exacerbate urban flooding or
even cause riverbank overflow. To investigate this process, the scenarios were re-simulated with the outfalls not connected
to the river. The pluvial flooding volumes without river influence is compared to the flooding volumes when the outfalls
were connected to the river, as shown in Fig. 14. It can be observed that under free outflow conditions, PF increases with
higher recurrence intervals. This suggests that, except for the outfalls directly influenced by the tide, pluvial flooding is
primarily driven by rainfall. For 1-h and 24-h durations, the flooding volumes during FP_S2, FP_S3, and FP_S4 stages
increase with higher recurrence intervals. For 3-h, 6-h, and 12-h durations, there are abrupt changes in flooding volumes
during FP_S2, FP_S3, and FP_S4 stages at recurrence intervals of 20-yr RP, 20-yr RP, and 5-yr RP, respectively. Before and
after these abrupt changes, the flooding volumes increase with higher recurrence intervals. Therefore, the trend of FP
corresponds to the flooding volumes during the FP_S2, FP_S3, and FP_S4 stages.



For 1-h, 3-h, 6-h, 12-h and 24-h durations, the maximum FP are 23.60%, 39.16%, 35.28%, 31.15%, and 35.70%, while the corresponding F_S3 percentages are 15.35%, 26.51%, 23.68%, 12.07%, and 19.14%. This indicates that riverbank flooding, even though it occurs at only a few nodes, can generate flooding volume close to 1/5 of the entire stage. The differences between FP and F_S3 are 8.25%, 12.65%, 11.60%, 19.08%, and 16.56%. It shows that about 10% of pluvial flooding is caused by the influence of river water level on the drainage outlets. As concluded above, fluvial flooding affects the

seriousness of urban inundation by influencing the elevation of outlets and causing direct riverbank flooding. The interaction between different flooding sources plays a significant role in exacerbating the inundation.

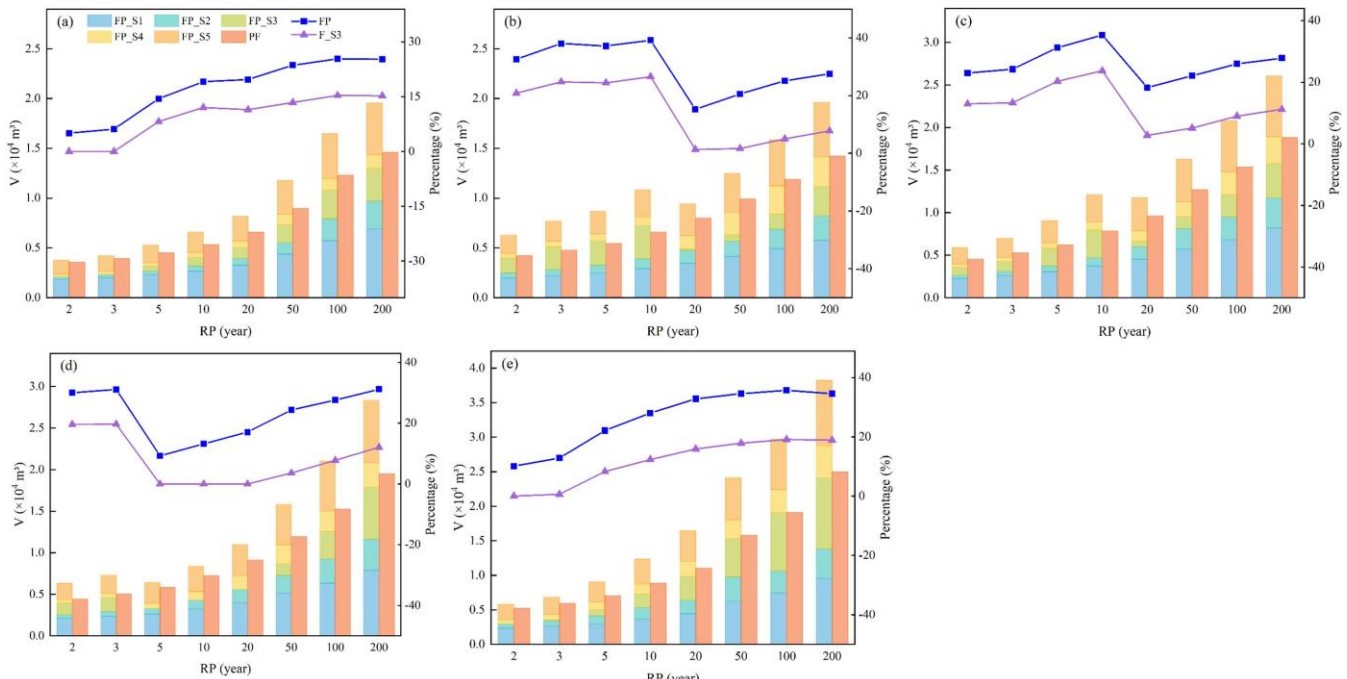

**Figure 14: Comparison of flooding volume with or without river: the sequence of duration from (a) to (e) is 1h, 3h, 6h, 12h and 24h.**

The impact degree indices at each stage (FP_S1, FP_S2, FP_S3, FP_S4, FP_S5) were calculated. Assuming only FP_S1 and FP_S5, the flooding volume is $m$, and the flooding volumes of each stage are as follows: $m/2$, 0, 0, 0, $m/2$. With the addition of stages FP_S2 and FP_S4, the flooding volumes for each stage are: $m_1$, $m_2/2$, 0, $m_2/2$, $m_3$. The results of the $D_t$ calculation are shown in Table 5. FP_S1 and FP_S5 are primarily influenced by rainfall, FP_S2 and FP_S4 are influenced by a combination of rainfall and other factors, and FP_S3 is mainly influenced by tidal levels. This indicates that tidal levels play

a significant role during the flood interaction stages.

**Table 5 Impact degree indices at different stages**

| $D_t$ | 1 h | 3 h | 6 h | 12 h | 24 h |
|---|---|---|---|---|---|
| FP_S1 | 0.92 | 0.94 | 0.94 | 0.97 | 0.89 |





| FP_S2 | 0.76 | 0.83 | 0.78 | 0.79 | 0.69 |
| FP_S3 | 0.48 | 0.55 | 0.49 | 0.34 | 0.38 |
| FP_S4 | 0.74 | 0.94 | 0.69 | 0.55 | 0.79 |
| FP_S5 | 0.94 | 0.97 | 0.96 | 0.95 | 0.88 |

**4.4 Causes and prevention measures of floods in drainage units**

This study provides an overview of the flooding causes in drainage units, considering both natural and societal factors. The natural factors include rainfall and tide, while societal factors primarily involve the drainage network, drainage outlets, and riverbank defenses. Drainage units DU1 to DU22 are influenced by both rainfall and the drainage network. DU14 to DU16 and DU18 to DU22 are affected by tide, while DU12, DU9, DU13, DU14, DUD15, U8, DU19, DU21, DU6, and DU16 are influenced by drainage outlets. DU15, DU16, DU19, and DU21 are additionally impacted by riverbank defenses. Summarizing the above, the causes of flooding can be categorized into five classes. Class I involves the joint action of rainfall and the drainage network. Class II comprises rainfall, the drainage network, and drainage outlets. Class III includes rainfall, tide levels, and the drainage network. Class IV consists of rainfall, tide levels, the drainage network, and drainage outlets. Class V extends Class IV by adding riverbank defenses. Classes I-IV represent the causes of pluvial flooding, while Class V represents the causes of general flooding. The classification of drainage units into these causal categories is presented in Fig. 15.

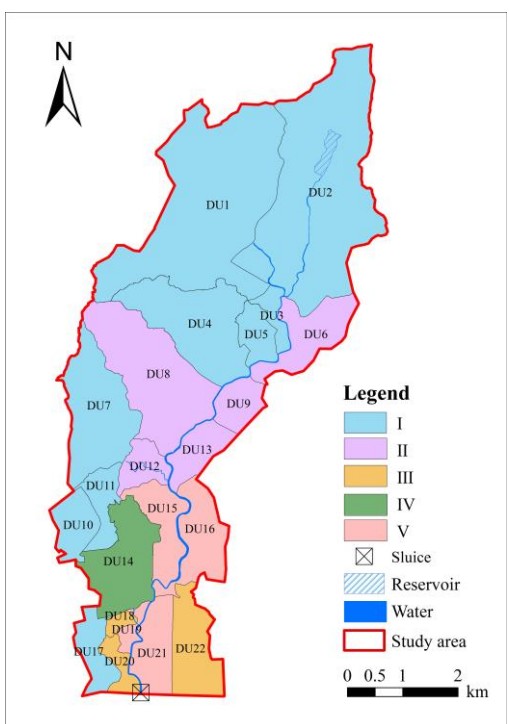

**Figure 15: Diagram of flood causes of drainage units.**



This research was conducted to explore the formation process and causes of flooding, with the aim of providing insights into effective flood prevention and management strategies. The results emphasize the multifactorial nature of flooding issues. It is recommended that urban areas should enhance their rainwater storage capacity and implement strategies that combine storage and drainage. This approach can help minimize peak outflow during floods, thereby mitigating the interactions between pluvial flooding and fluvial flooding. This is significant for addressing various causes of flooding. In the drainage units characterized by causes of Class II, Class IV, and Class V, there are instances of inappropriate drainage outlet elevations. Class V causes also can identify the nodes in the midstream areas that are prone to river embankment breaches. Based on these findings, it is advisable to study river and drainage outlet design standards that are more suitable for current and future climate conditions. Additionally, it's essential to explore how ecological engineering can improve the performance of rivers and drainage outlets. Management of drainage outlets should also be a focal point. During high tide periods, drainage systems in the units associated with causes of Class III to Class V may experience operational challenges. Therefore, the consideration of stronger drainage facilities to alleviate flood risks is recommended. For drainage outlets directly affected by tidal buoyancy, such as DU22, planning for protective measures like embankments or other protective structures can help reduce the impact of tides. Moreover, for the units with significant interaction intensity, such as DU12, targeted flood mitigation measures should be implemented to enhance the flood resistance of the entire drainage system.

## 5 Discussion

Previous studies have primarily focused on analyzing urban pluvial flood risks triggered by heavy rainfall (Zhang et al., 2017; Zhang et al., 2021; Zou et al., 2022). These studies have been effective under specific boundary conditions. However, for coastal cities, storm surges are also a critical influencing factor. With the combined effects of rainstorm and storm surges, flooding can manifest not only as urban pluvial flooding but also in other forms. Therefore, it is essential to conduct an in-depth examination of the multifaceted flood risks driven by rainfall and storm surges to ensure research findings align more closely with real-world scenarios. This study introduces the concept of the impact degree index, which quantitatively assesses the impact of rainfall and tide processes of various durations on flooding. Simultaneously, it takes into account the interactions between fluvial flooding and urban pluvial flooding, revealing the exacerbating effects of flood interactions on flood disasters. Through this analysis, a comprehensive understanding of the causes of flooding in drainage units is obtained, which is helpful in making targeted flood prevention and management measures. This holistic analysis not only helps overcome potential limitations of previous research but also provides a new perspective for further investigating the complexities of flooding. As a result, it contributes to improving flood management and response strategies.

The results of this study indicate that under free outflow conditions at drainage outlets, the pluvial flooding volume for 1-h rainfall events of 100-yr and 200-yr RP is greater than that for 3-h events of 100-yr and 200-yr RP. Additionally, the 6-h rainfall events of 2-yr to 100-yr RP have larger pluvial flooding volumes than the 12-h events of 2-yr to 100-yr RP. This





demonstrates that the rainfall pattern significantly influences flooding. Specifically, a higher average rainfall intensity and larger peak coefficient lead to greater pluvial flooding volume. This finding is supported by various studies(Cao et al., 2021; Jiang and Yu, 2022; Palla et al., 2018). Higher rainfall intensity tends to result in more rainfall being concentrated and released in a shorter period, making drainage systems more prone to turning saturated, thus increasing the likelihood of flooding. The impact of rainfall and tide processes on flooding is also reflected in the peak timing. Previous research has focused on the influence of rainfall peak timing on the flood process (Chen et al., 2023). Chen et al. found that later rainfall peak timing increases flood risk, but with an increase in return period, the impact of rainfall peak timing decreases (Chen et al., 2018). Similar conclusions have been drawn by Cheng et al., stating that floods with shorter time lags and later peak timings intensify the impact of flooding (Cheng et al., 2020). This study also highlights that under the same recurrence interval, rainfall events with larger peak timings are more destructive than those with earlier peak timings. In areas affected by tides, the difference between the peak timing of rainfall and tide processes significantly impacts flood risk (Zheng et al., 2013). Shen et al. (2019) suggests that the most severe floods occur when the time lag is between -1 to 2 hours Therefore, in the design of drainage systems and flood control strategies, it is essential to comprehensively consider both rainfall and tide processes rather than relying solely on the recurrence interval as an assessment metric. Despite differences in rainfall peak timing, rainfall, and tide peak timings leading to variations in calculated flooding volumes, the impact degree index is not affected by these differences.

The flooding issues involves numerous complex factors, including climate, topography, land cover, urbanization levels, human activities, and more (Liu et al., 2021; Zhang et al., 2020). Merz et al. pointed out in their research that heavy rainfall or prolonged precipitation is often a driving factor for extreme floods, but the previous state of the watershed also has a significant impact on floods (Merz et al., 2021).In low-lying coastal areas, river flooding occurring simultaneously with high tides increases the severity of floods. Iya suggests that flooding has various causes, such as improper drainage systems, pollution, urban management, environmental factors, weather, dam failures, among other factors (Durumin Iya, 2014) This study encompasses multiple aspects of urban flooding causes, including rainfall, storm surges, drainage systems. However, when examining flooding issues at a microscale, there are more specific details and particular situations that need attention. For instance, clogged or debris-laden stormwater grates can affect the normal discharge of rainwater, potentially leading to localized flooding issues. Aging or damaged drainage pipes may risk leakage or collapse, hindering the flow of water and causing flooding (Mohandes et al., 2022). Future research can further explore the relationships between topography, land cover, and other factors in relation to flooding causes, providing a more comprehensive understanding of the factors contributing to flooding.

## 6 Conclusion

This study provides a quantitative assessment of the impact of rainfall and tide levels on flooding and analyzes the interactions between fluvial flooding and pluvial flooding. It classifies flood-prone areas into different regions based on the


primary natural factors (rainfall and storm surges) and social factors (drainage systems, outfalls, and riverbank). This classification helps decision-makers identify the causes of flooding in various drainage units. The main conclusions are as follows:

(1) For 1-h, 3-h, 6-h, 12-h, and 24-h durations, the most suitable joint distribution functions for rainfall and tide levels are Gumbel Copula, Frank Copula, Gaussian Copula, Gaussian Copula, and Frank Copula, respectively. The $R_{Kendall}$ falls between the $R_{Or}$ and the $R_{And}$, thereby avoiding overestimation or underestimation of the risk areas. Using the MPWF method, reasonable design values of rainfall and tidal levels were obtained.

(2) Based on the stormwater flood model, the flooding volume generally increases with the increase in rainfall total amount and tide peak value. Flooding in the study area is influenced by both rainfall and tide, with rainfall being the primary driver. The influence of tide on flooding decreases with increasing distance from the drainage units to Shahe River and Pearl River in spatial terms, and it is most significant in the 24-h duration.

(3) The maximum proportion of the flooding volume due to the water topping the outlets and the proportion of flooding volume due to riverbank overflowing is 19.08% and 26.51%, respectively. It shows that the interaction of different types of floods exacerbates flood disasters. The influence of tides on floods mainly occurs during the interaction between fluvial flooding and pluvial flooding.

**Data availability**

"Technical Report on the Compilation of Guangzhou Rainstorm Intensity Formula and Design Rainfall Patterns" is publicly available on the Guangzhou Water Authority (http://swj.gz.gov.cn/gkmlpt/content/8/8835/mpost_8835843.html?eqid=b613570d0000e49000000006648a743f#1052).

**Competing interests**

The authors declare that they have no conflict of interest.

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
