# Peer review of "Exploring the driving factors of compound flood severity in coastal cities: a comprehensive analytical approach"

_Hydrology and Earth System Sciences, 2024_

## Author Comment (AC1)

**Response to Anonymous Referees**

**Dear Editor,**

Thank you very much for the constructive feedback from you and the reviewers. We have addressed all the reviewers' comments. Please find a point-by-point reply below.

Kind regards

Ting Zhang

**Response to First Referee**

In this study, the authors conducted a comprehensive investigation of the driving factors behind compound flooding in coastal cities using a combination of hydrodynamic modeling and mathematical statistics. Their research yielded insights into the impact of rainfall and tidal levels on compound flooding, as well as the contributions of different types of floods to compound flooding. The conclusions of this study are novel and have positive implications for risk management. The chosen topic also aligns with the scope of the journal "Hydrology and Earth System Sciences.", I have several suggestions and recommendations that would help enhance the manuscript.

**Specific comments**

1. It is necessary to indicate whether there are tide prevention facilities in the study area and how this factor is considered in the model.

**Response:** Thank you for your suggestion. Indeed, the issue of flood control facilities is a significant factor in our research. A tidal gate has been installed at the river outlet in the study area. We will include detailed descriptions of the flood control facilities and model settings in the model construction section. Specifically, a tidal gate with a total net width of 24 meters is installed at the river outlet. In the model, the design tide level is set outside the gate. The operational rule is set such that when the tide is receding and the water level inside the gate is higher than the water level outside the gate, the gate is opened to discharge water. The gate status is checked every hour to determine if adjustments are needed.

2. In the "Model construction and validation" section, a one-dimensional and two-dimensional hydrodynamic coupling model was constructed, but the analysis of flood severity primarily relied on flood volume, with limited analysis on indicators such as flood area and flood depth. This aspect needs to be supplemented.

**Response:** Thank you for your feedback. We will include a detailed analysis of flood area and flood depth in the results section. Although using flood area or flood depth alone is not suitable for calculating impact index, these data are indeed necessary for comprehensive analysis to provide a more thorough assessment of compound flood impact.

3. The paper only mentioned the length of the original data but did not elaborate on the sample data situation for constructing the Copula function.

**Response:** Thank you for your suggestion. We will include details on sample selection methods and specific data in the methodology section. Specifically, we calculate the maximum values of rainfall for different durations (1h, 3h, 6h, 12h, 24h) each year, using the minimum value in the set of maxima as the threshold. We select rainfall events exceeding this threshold. If there are multiple events exceeding the threshold in a single day, only the largest one is chosen. For each selected rainfall sample P, the highest tide level value Z on the same day is identified as the tidal sample. In this study, the rainfall thresholds for 1h, 3h, 6h, 12h, and 24h are 36mm, 56mm, 58mm, 66mm, and 78mm, respectively. The final number of samples for each duration is 48, 39, 49, 49, and 52, respectively.

4. The paper is lengthy, and to be concise, compressing the research methods section would be helpful. For example, common formulas in Correlation and Copula.

**Response:** Thank you very much for your suggestion. We will reorganize the methodology section based on your advice. For example, commonly used correlation formulas will be provided through citations rather than detailed explanations in the text.

5. In the section "Spatial interaction of drainage units," the analysis of the interaction forces between different drainage units is not highly relevant to the main theme of this paper.

**Response:** We appreciate your suggestion. The original aim was to identify the key areas for flood control and disaster mitigation management by analyzing the hydraulic connections of different drainage units. We will consider reorganizing this section or removing it from the study based on the comments from multiple reviewers and the editor.

6. In section 4.4 "Causes and prevention measures of floods in drainage units," flood prevention measures should not be discussed in the research results. It is suggested to elaborate on them in the discussion section.

**Response:** Thank you very much for your suggestion. We have moved the discussion of flood control measures to the discussion section, where we will elaborate on specific prevention and control measures for different causes of flooding.

7. The conclusion needs further refinement.

**Response:** We will further refine and summarize the conclusions of this study.

**Technical comments**

L11: The word "Currently" is repeated.

**Response:** Thank you for your feedback. The errors have been corrected.

L31: Change "in this year" to a specific year.

**Response:** Following the suggestion, the date has been revised to 2023.

Figure 1: Mark all drainage outlets in the figure.

**Response:**We will revise Figure 1 to include all drainage outlets.

Table 1: Explain why the RMSE of the edge distribution function corresponding to the optimal tidal level for 3h is not the best.

**Response:** For the 3-h duration, although the RMSE of the edge distribution function corresponding to the optimal tide level is not ideal, the difference from the minimum value is minimal. It is common to encounter this phenomenon when using multiple evaluation metrics. It is necessary to consider the research question and select the main evaluation indicator. In this study, we prioritize AIC as the main evaluation indicator. Relevant explanations will be added to the methodology section.

L430-431: What is the specific relationship? It needs to be clarified.

**Response:** In the discussion section, we explore the relationship between total rainfall, rainfall peak, peak time, and the severity of flooding. Specifically, we observe that higher average rainfall intensity and peak coefficient lead to larger flood volumes. Rainfall events with larger peak values tend to be more destructive compared to those with earlier peak times. We will further elucidate this relationship by considering combined scenarios of rainfall and tide level.

L457-459: Drainage unit 14 appears in both cases simultaneously, please verify.

**Response:** It has been verified that the Dt of drainage unit 14 is close to 1. L458-460 has been modified as follows: "For DU15 to DU16 and DU18 to DU22 in Fig. 12(b), Dt values are around 0.5, which indicates that flooding is the result of the 460 combined effects of rainfall and tides."

L460-461: Same as above.

**Response:** This part has been modified as follows: "In spatial terms, the Dt values for DU15 to DU16 and DU18 are greater than those for DU19 to DU22 ."

L508-509: This is an observation, not a conclusion.

**Response:** The sentence can be rephrased as "The trend of FP corresponds to the total flooding volumes of the FP_S2, FP_S3, and FP_S4 stages."

Figure 14: The discussion of the causes of sudden changes is crucial.

**Response:** Thank you for your feedback. The main reason is the influence of the tidal gate control, which we will delve into in the discussion section.

L520-522: This should be in the Methods section.

**Response:** Thank you very much for your feedback. We will move this content to Section 3.6.

---

## Author Comment (AC3)

Response to Anonymous Referees

Dear Editor,

Thank you very much for the constructive feedback from you and the reviewers. We have addressed all the reviewers' comments. Please find a point-by-point reply below.

Kind regards

Ting Zhang

Response to Referee #2

In this study, the authors thoroughly explored the factors driving compound flooding in an urban area by combining hydrodynamic modeling and multivariate statistics. Their research provided insights into the analysis of the relative contribution of rainfall and tidal levels on compound flooding and the roles of different flood scenarios. The study is fairly novel and findings are beneficial for flood risk management and applications. Further, the scope and findings of the study align with the scope of the journal "Hydrology and Earth System Sciences," and I recommend its publication but after some revisions. Below, the authors can find my suggestions and recommendations, which I believe would contribute to improving the manuscript and clarifying the methodology.

**Major Comments:**

1. Copulas and marginal distributions are fitted to the "tidal levels." Is this just the astronomical tide, or do these values include wind-driven surge (in the case of Typhoon/Cyclone conditions) and river discharge components as well? The authors mention "storm surge" as a critical factor for compound flooding several times (L346, L456, L564), but it's not clear how storm surge is taken into account when modeling "tidal levels" in this study. If the tidal range is higher and the storm surge component is relatively very small (or if only the astronomical tide is considered), fitting an extreme value model might be fundamentally incorrect since the astronomical tide is deterministic. Therefore, a clear explanation of the terminology is needed. Furthermore, are the tidal boundary conditions used at the sluice gate (as outflow boundary conditions?)? The authors mention a tidal gate, but the location is not shown in the figure.

**Response:** Thank you very much for your meticulous review and valuable comments. You raised the issue regarding the fitting of extreme values for "tidal levels," particularly how to account for storm surge factors. In this study, "tidal levels" is not solely based on astronomical tides but also includes the water level rise caused by

wind-driven surge and river discharge. The study area experiences abundant rainfall and frequent typhoons, leading to significant tidal level increases due to wind-driven surge and discharge from heavy rainfall. For example, Typhoon Mangkhut in 2018 resulted in the highest tidal level recorded at the tide gauge in this study, reaching 3.28 m. Therefore, the tidal level is not deterministic. We will provide a clear explanation of the terminology in the revised manuscript.

Regarding the issue of sluice gate and tidal boundary conditions, as you correctly pointed out, the sluice gate uses tidal level as the outflow boundary condition. This setup ensures that the model can account for tidal influences. The tidal gate in this paper is the sluice gate. The location of the sluice gate is shown in Figure 1. We will unify the terminology of pictures and words.

2. When selecting the sample for joint probability analysis, the tidal level is chosen as the "highest tidal level of the corresponding day." This method implies that the calculated tidal design levels could occur within a 24-hour period from the rainfall event. In the numerical modeling process, a single event is used as a representative for the tidal levels process, but the timing of the rainfall peak and tidal peaks in generating design scenarios is not explained. Additionally, figure 8(a) shows the peak rainfall and peak tide almost coinciding, which is rare in reality (not the most probable 200-year event). The 3-hour rainfall duration shows the peak rainfall occurring approximately half an hour after the peak tide, whereas the 12-hour rainfall duration shows the peak rainfall occurring approximately 3 hours before the peak tide. How are the corresponding time lags between peak rainfall and peak tide determined for each scenario? Are they coming from a probabilistic basis? This timing significantly impacts the resultant flood volume and might be the main reason for the variations discussed around L430.

**Response:** Thank you for your detailed review of our research methodology and valuable suggestions. Our study selected the tidal level on May 22, 2020, as the typical tidal level process, based on the "Comprehensive Plan for Drainage (Rainwater) and Flood Prevention in Guangzhou (2022-2035)." The peak time of the typical tidal level is close to the peak time of the 24-hour design rainfall. For tide levels with rainfall durations of 1 h, 3 h, 6 h, and 12 h, this study designed the tidal levels with the same rising and ebb tide times, with the peak tide level occurring in the middle of the simulation period. The corresponding explanation will be supplemented in the revised manuscript. In future work, we will consider increasing the lag time and establishing the joint distribution of the three variables to design rainfall and tide level combinations. Additionally, we will discuss in detail about the impact of the lag time between peak rainfall and peak tidal level on flood volume in the revised manuscript and explain the reasons for the variations in L430.

3. 41 and 42 are used to evaluate the spatial interactions between drainage units (DUs). The equations only rely on the masses and distances of two drainage units. To clarify the conclusions around L445-L450, it is essential to understand what the

spatial interaction force represents. For example, the spatial interaction force between DU22 and DU6 (which are far away from each other) is 0.75. Does this relatively higher value suggest a higher level of interconnectedness or influence on each other, than even adjacent drainage units? Please explain. Also, it is not clear how the spatial interaction of the drainage units relates to the main objectives of the study.

**Response:** Thank you very much for your meticulous review of the spatial interaction between drainage units in our study. The primary aim of this research is to analyze the sources of compound flooding. The study on the spatial interaction of drainage units seeks to provide a scientific basis for identifying key areas for flood management. The spatial interaction between drainage units is based on the urban gravity model, which posits that the intensity of interaction between cities is proportional to their size (mass) and inversely proportional to their distance. In this study, the spatial interaction between drainage units is analogous to the interaction between cities. Spatial interaction force represents the strength of the connection between drainage units.

Theoretically, the spatial interaction force between DU22 and DU6 is relatively high, indicating a higher level of interconnectedness compared to adjacent drainage units. Upon careful examination of the calculation process, we found that the drainage pipeline length of DU22 is significantly greater than that of other drainage units on the left bank, resulting in a higher spatial interaction force between DU22 and other drainage units, even with more distant to DU6. This is related to the simplification of the model assumptions. In this study, the mass indicators only included flood volume and drainage pipeline length, which may lead to an incomplete interpretation of mass. Additionally, the weights of mass and distance are equal, which could introduce uncertainty. Future work will involve conducting sensitivity analyses to select more mass indicators and set indicator weights to reduce uncertainty in the results. We will supplement the corresponding explanations and analyses in the revised manuscript.

4. It is necessary to provide additional details on the types of data used in the study. For instance, specify whether rainfall data is gridded or gauge data, and if gauge data is used, indicate the number of gauges involved. What are the locations of the tidal data and rainfall used? If so, do you assume a uniformly distributed rainfall over the entire catchment?

**Response:** Thank you very much for your meticulous review and attention to the data types. The rainfall and tide level data used in this study are gauge data. We obtained rainfall data from 3 gauges and tidal data from 1 gauge. We assumed that the distribution of rainfall is uniform across the entire catchment area, so we used the average value from the 3 gauges. This assumption is based on the representativeness of the gauges and the limitations in obtaining long-sequence data. Future work will consider using data from more gauges or high-resolution gridded data to improve the accuracy and reliability of the model. We will include detailed descriptions of the data in the revised manuscript and indicate the specific locations of these gauges in the figures.

5. The paper lacks a proper discussion about the main assumptions and limitations of the process. For instance, using only 16 years of data to estimate a 200-year design event introduces significant uncertainty in modeling the dependence structure and the tail distributions. Since the study proposes a "universal" method, it is crucial for readers to understand these assumptions and uncertainties, especially when the method is to be applied to other study sites.

**Response:** Thank you very much for your suggestion. We will add a section in the revised manuscript to discuss in detail the assumptions and limitations of this study. This will be very helpful in enhancing the rigor and applicability of the "universal" method presented in this paper.

Minor Comments

1. L11 "currently" repeats.

**Response:** Thank you for pointing out the repetition error in the text. We will remove the redundant "currently" in the revised manuscript.

2. In the abstract, the authors mention the Kendall return period for the combined event of rainfall and tidal levels, greater than the "Or" return period and less than the "And" return period. The Kendall return period is typically expected to fall between the "Or" and "And" scenarios for combined events. It's not a significant finding of the study to be mentioned in the abstract.

**Response:** Thank you for your meticulous review of the abstract content. Based on your suggestion, we will remove the description of the Kendall return period from the abstract.

3. L32, and L36, could you provide references?

**Response:** Thank you very much for your review comments. We have supplemented the relevant references as per your suggestions to enhance the accuracy and reliability of the paper. In response to your request, we have added the following references:

Dorrington, J., Wenta, M., Grazzini, F., Magnusson, L., Vitart, F., and Grams, C.: Precursors and pathways: Dynamically informed extreme event forecasting demonstrated on the historic Emilia-Romagna 2023 flood, EGUsphere, 2024, 1-27, https://doi.org/10.5194/egusphere-2024-415, 2024.

Jiao, Z., Zhang, Z., and Wu, L.: SAR-based dynamic information retrieving of the Beijing-Tianjin-Hebei flood-inundation happened in July 2023, North China, Geomatics, Natural Hazards and Risk, 15, 2366361, https://doi.org/10.1080/19475705.2024.2366361, 2024.

Marengo, J. A., Cunha, A. P., Seluchi, M. E., Camarinha, P. I., Dolif, G., Sperling, V. B., Alcântara, E. H., Ramos, A. M., Andrade, M. M., and Stabile, R. A.: Heavy rains and hydrogeological disasters on February 18th–19th, 2023, in the city of São Sebastião, São Paulo, Brazil: from meteorological causes to early warnings, Nat. Hazards, 120, 7997-8024, https://doi.org/10.1007/s11069-024-06558-5, 2024.

4. Figure 2: what represents the purple arrow from correlation analysis to "effects of social factors on flood"?

**Response:** Thank you very much for your meticulous review. We indeed recognize the error in the direction of the arrow and apologize for it. The original intention was to express that the analysis of "effects of social factors on flood" is based on the combination of rainfall and tide level events. We have made the necessary corrections to Figure 2 to ensure the arrows point correctly.

5. L61, in underdeveloped areas, fluvial flooding may indeed be more prevalent due to the natural terrain and lack of infrastructure to mitigate such events. However, it is not accurate to say that "only" fluvial flooding exists in these areas. Could you explain or provide references?

**Response:** Thank you for your correction. The statement in L61 is indeed inaccurate. In underdeveloped areas, fluvial flooding may be more common. However, it is inappropriate to simplify the flooding issues in these regions as "only fluvial flooding." These areas may also encounter different types of flooding events caused by factors such as tide. We will re-examine this paragraph and provide a more accurate description in the revised manuscript.

6. Provide references for the "Pilgrim & Cordery rainfall model", "Maximum Possible Weighting Function" and "Pilgrim & Cordery rainfall model".

**Response:** Thank you very much for your suggestions regarding the references. We have supplemented the corresponding references for the "Pilgrim & Cordery rainfall model" and the "Maximum Possible Weighting Function."

For the "Pilgrim & Cordery rainfall model," we will cite the following references:Pilgrim, D. H. and Cordery, I.: Rainfall temporal patterns for design floods, Journal of the Hydraulics Division, 101, 81-95, https://doi.org/10.1061/JYCEAJ.0004197, 1975.

For the "Maximum Possible Weighting Function," we will cite the following reference:Gibson, S., Wills, A., and Ninness, B.: Maximum-likelihood parameter estimation of bilinear systems, IEEE Transactions on Automatic Control, 50, 1581-1596, https://doi.org/10.1109/TAC.2005.856664, 2005.

7. Figure 6 seems incorrect. The theoretical distributions (GEV, Gamma) should be smooth unless you don't change the distribution parameters.

**Response:** Thank you very much for pointing out the issue in Figure 6, and we apologize for the oversight. Upon careful examination, we found that the legends for the empirical distribution curve and the theoretical distribution curve were reversed, causing confusion. We have corrected Figure 6 to ensure that all legends accurately reflect their corresponding distribution types. The theoretical distribution curve is now smooth and will be presented in the revised manuscript.

8. What types of information does Figure 7 (b) provide? What are the insights we can gain into joint distribution probabilities? The color scale is also not given.

**Response:** Thank you very much for your detailed review of Figure 7(b). We apologize for not providing complete information. In Figure 7(b), we can see the joint probability of two variables for any combination. The graph shows that a significant portion of the joint probability is below 0.5, suggesting that the joint probability of rainfall and tide level has a sharp peak and heavy tail characteristic. This indicates that a majority of the data is concentrated within a specific range, while the rest of the data is spread out across a wide range of intervals. We will provide a detailed analysis of Figure 7(b) in the revised manuscript. Additionally, you correctly pointed out the absence of a color scale. The color scale is a crucial element for understanding the data distribution in the figure. We have modified Figure 7(b) to include the color scale, which will be presented in the revised manuscript.

9. Although the peak tidal levels could be negative (depending on the datum used), the gamma distribution is selected to fit the peak tidal levels for some rainfall durations. The gamma distribution is lower bounded by zero thus making it impossible to sample negative realizations from the fitted distribution (if necessary).

**Response:** Your opinion is indeed correct. The lower bound of the gamma distribution is zero, making it unsuitable for fitting data that may include negative values. In our study, the peak tidal levels are all positive, based on the reference datum we selected. Therefore, using the gamma distribution in this context is appropriate. We will include an explanation of this point in the revised manuscript to help readers better understand the conditions under which the gamma distribution is used.

10. The extreme sample selection is not well explained. Does it consider annual maxima? To ensure that events are independent, it is important to make sure that the outcome of one event does not affect the outcome of another. This can be achieved by using random sampling techniques and ensuring that each event is selected independently of the others. Additionally, it is important to have a sufficiently large sample size in order to make accurate inferences about the population. The sample

size should be determined based on the desired level of confidence and the variability of the data.

**Response:** Thank you very much for your attention and suggestions regarding the sample selection method. In this study, we considered the annual maximum values. First, we calculated the maximum rainfall for each year, forming set D, and used the minimum value in set D as the sample threshold. Then, from 16 years of historical data, we selected events greater than or equal to this threshold as preliminary rainfall samples. Following the principle of the minimum inter-event time of 6 hours between two samples, if the minimum inter-event time was less than 6 hours, we selected the larger of the two samples. Furthermore, if multiple events within a single day exceeded the threshold, we chose only the largest one to generate the final rainfall samples. For each selected rainfall sample P, we identified the highest tide level value Z on the same day as the tide sample. In this study, the rainfall thresholds for 1-hour, 3-hour, 6-hour, 12-hour, and 24-hour durations were 36 mm, 56 mm, 58 mm, 66 mm, and 78 mm, respectively. The final number of samples for each duration was 48, 39, 49, 49, and 52, respectively. A more detailed explanation will be provided in the methodology section of the revised manuscript.

11. Figure 10: The study area is in the legend (red). But cannot be seen in the figure.

**Response:** Thank you very much for pointing out the issue in Figure 10, and we apologize for the oversight. Upon careful examination, we found that the color annotation for the study area was indeed incorrect. The red marker you mentioned was not correctly displayed in the figure, which may have caused difficulties in understanding. We have corrected Figure 10 to ensure that the color of the study area in the legend matches the color displayed in the figure.

12. Figure 12: Do these different colors represent any information? If not use a single color.

**Response:** Thank you for your meticulous review of the color usage in Figure 12. Indeed, the different colors used in Figure 12 did not represent specific information, which may have caused unnecessary confusion. Based on your suggestion, we have modified these colors to a single color to make the graphical representation more concise and clear.

13. It's not clear how this observation is made. Despite differences in rainfall peak timing, rainfall, and tide peak timings leading to variations in calculated flooding volumes, the impact degree index is not affected by these differences.

**Response:** Thank you very much for your attention to our research results. Regarding the observation you mentioned, "Despite differences in rainfall peak timing, rainfall, and tide peak timings leading to variations in calculated flooding volumes, the impact degree index is not affected by these differences," we recognize that there was a

writing error in the original text, and we apologize for this. We will reorganize and refine this section in the revised manuscript.

14. It's not clear how this conclusion is made: "This study also highlights that under the same recurrence interval, rainfall events with larger peak timings are more destructive than those with earlier peak timings"? Have you run many events for the same return period by only changing the peak timing?

**Response:** Thank you for your meticulous review. Regarding your mention: "This study also highlights that under the same recurrence interval, rainfall events with later peak timings are more destructive than those with earlier peak timings," and its previous sentence "Similar conclusions have been drawn by Cheng et al., stating that floods with shorter time lags and later peak timings intensify the impact of flooding" are all derived from the work of Cheng et al. (Cheng, T., Xu, Z., Yang, H., Hong, S., and Leitao Joao, P.: Analysis of Effect of Rainfall Patterns on Urban Flood Process by Coupled Hydrological and Hydrodynamic Modeling, Journal of Hydrologic Engineering, 25, 04019061, 10.1061/(ASCE)HE.1943-5584.0001867, 2020.).We will rewrite these sentences in the revised manuscript to clearly indicate that they are conclusions from other researchers and to cite the original source correctly.

15. The authors suggest a set of equations (from eq. 30 to eq. 40), but poorly explained. Consider elaborating more about the governing process of equations. Additionally, it's mentioned that "calculate the range of rainfall and tidal level design values for different durations from 2-yr RP to 200-yr RP using the following formula" (L286,), but the formula doesn't give the range of rainfall, and since it divides the range by 200yr Rainfall value. So, $\Delta X_t$ will be a dimensionless parameter that is related to the amount of variability (range) of rainfall. Same for Tides. Same in L304, $\Delta Vx_t$ and $\Delta Vy_t$ do not quantify the variation of flooding volume. Accordingly, check $Dx_t$ and $Dy_t$.

**Response:** Thank you for your detailed review of the equations and related calculation methods in our paper. We acknowledge that our explanation and description in this section were indeed insufficient, and we apologize for this. We will rephrase the control process of the equation set in the revised manuscript to help readers better understand the physical significance and application value of these equations.

Regarding your mention of L286, "calculate the range of rainfall and tidal level design values for different durations from 2-yr RP to 200-yr RP using the following formula." The original intention was to express "calculating the variation range of rainfall and tide levels from 2-year to 200-year rainfall and tide levels," where the rainfall and tide levels are derived from the design values in Section 3.3. Our incorrect expression misled your understanding, and we apologize for this. We will correct this statement and re-examine $\Delta Vx_t$, $\Delta Vy_t$, $Dx_t$, and $Dy_t$ to ensure that all parameters and variables are clearly and correctly defined.

16. The paragraph from L543 to 558 is more about discussing the results. Consider moving it into the discussion section.

**Response:** Thank you very much for your suggestion, which is very helpful for improving the logical flow and structure of our paper. We will adjust the placement of this section to ensure that the overall structure of the paper is clearer and more coherent.

17. The paper is lengthy. Consider moving some of the extra results to the supplementary materials. For example, Tables 1 and 2. Commonly used equations, such as those for correlation and distributions, can be omitted from the text. Instead, it is sufficient to cite the relevant references for these well-known equations.

**Response:** Thank you very much for your suggestion, which is extremely beneficial for optimizing the length and organization of our paper. We will make adjustments and refinements to the manuscript as per your advice.

18.Check the way of citing publications in the manuscript according to the HESS guideline (e.g., L40, L42, L50).

**Response:** Thank you for pointing out the issue with the citation format. We will thoroughly check and revise the references to ensure they strictly adhere to the HESS guidelines.

---

## Author Response (AR1)

Dear Editors and Reviewers:

Firstly, I would like to thank you for taking the time to review our manuscript and for providing many valuable suggestions and comments. We have carefully considered all your comments and made corresponding revisions to the manuscript. Below are our responses to each of your comments. Throughout the response, the editors' comments reviewers' comments are presented in black, and our responses are in blue. The line in red correspond to those in the clean version of the revised manuscript.

Kind regards

Ting Zhang

**Response to Editors**

We have now received two reviews of your manuscript 'Exploring the driving factors of compound flood severity in coastal cities: a comprehensive analytical approach'. The reviewers and the editorial team feel that the topic of the article is an appropriate fit for Hydrology and Earth System Sciences and that the work merits publication and that the methods and results contribute to the literature on the topic of compound flooding. However, the reviewers have also expressed that revisions to the text are necessary in order to improve the content of the manuscript prior to publication. Specifically, several comments address a lack of clarity around the terminology that was used to describe the variables used and how they were measured. In addition, the reviewers find the paper lengthy and have provided several recommendations on how the manuscript can be shortened and where information that is widely available elsewhere in the literature (e.g., common equations) can be removed to streamline the paper. In your response to the reviewer comments, please clearly indicate for each comment how you have (or have not) changed the manuscript to address it.

Response to the editor's comment "Specifically, several comments address a lack of clarity around the terminology that was used to describe the variables used and how they were measured.":

Thank you very much for your valuable suggestions. We have realized that our explanations regarding storm surges, spatial interaction forces, impact index, and the types of rainfall and tide level data were not sufficiently clear. We apologize for this. In

the revised manuscript, we have made comprehensive revisions to enhance the rigor of the paper. The specific revisions are as follows:

**(1) Regarding storm surge and tide level**

In this study, the tide levels used for extreme value fitting are not only based on astronomical tides but also include water level rises caused by wind-driven surges and river flows. The tide level data is sourced from gauge data and reflects the influence of the above factors. We have supplemented the explanation of terms and revised the description of storm surges in the paper according to your suggestions:

1) Tide levels include not only astronomical tides but are also influenced by meteorological tides, with storm surges being an extreme form of meteorological tides. (lines 35-36)

2) The tides of the Pearl River are influenced not only by astronomical tides but also significantly by storm surges and upstream river flood flows. (lines 94-96)

3) The tide level data reflects the combined effects of astronomical tides, storm surges, and river flows. During Typhoon Mangkhut in 2018, the tide level gauge in this study recorded the highest tide level (3.28 m). (lines 106-107)

4) Changed "storm surge" to "tide level". (lines 38, 53)

**(2) Regarding the spatial interaction force**

Spatial interaction force represents the connection strength between drainage units. A higher interaction value indicates a stronger spatial influence of the drainage unit, warranting more attention in flood management. Relevant explanations have been added to Section 3.5:

Flooding in each drainage unit is influenced through various pathways, including pipes, rivers, and surface routes. Identifying key drainage units most closely linked with others is crucial for effectively formulating flood management strategies. This study draws on the principles of the urban gravity model to quantify the interaction force between drainage units (Zhao et al., 2021). Spatial interaction force represents the connection strength between drainage units. A higher interaction value indicates a stronger spatial influence of the drainage unit, warranting more attention in flood management. (lines 292-296)

**(3) Regarding the variables in the calculation of the impact index**

The impact index is calculated using a set of equations. We have reorganized Section 3.4 to ensure that all variables are properly defined. The specific revisions are as follows:

The impact index of rainfall on flooding ($D_{x,t}$) is calculated by comparing the impact of rainfall on flooding with the total impact of both rainfall and tide levels, as shown in Eq. (4). Similarly, the impact index of tide levels on flooding ($D_{y,t}$) is calculated by comparing the impact of tide levels on flooding with the total impact of both rainfall and tide levels, as shown in Eq. (5). In this study, the impact index of tide levels on flooding can also be calculated as $1 - D_{x,t}$. The value of $D_{x,t}$ ranges from 0 to 1, where 0 indicates that flooding is solely influenced by tide levels, and 1 indicates that flooding is solely influenced by rainfall. A larger $D_{x,t}$ value signifies a more significant impact of rainfall on flooding, with a relatively weaker influence of tide levels.

$$D_{x,t} = \frac{F_{x,t}}{F_{x,t} + F_{y,t}} \tag{4}$$

$$D_{y,t} = \frac{F_{y,t}}{F_{x,t} + F_{y,t}} = 1 - D_{x,t} \tag{5}$$

In the above equations, $t$ represents different durations. The impact of rainfall on flooding, $F_{x,t}$, refers to the flood volume change due to the rainfall change. The impact of tide levels on flooding, $F_{y,t}$, refers to the flood volume change due to the tide levels change. The calculation formulas are as follows:

$$F_{x,t} = \frac{\Delta V_{x,t}}{\Delta x_t} \tag{6}$$

$$F_{y,t} = \frac{\Delta V_{y,t}}{\Delta y_t} \tag{7}$$

In the above equations, the change in rainfall, $\Delta x_t$, is the relative change from 2-yr return period to 200-yr return period, and the change in tide levels, $\Delta y_t$, is also the relative change from 2-yr return period to 200-yr return period. The calculation formulas are as follows:

$$\Delta x_t = \frac{X_{2,t} - X_{1,t}}{X_{2,t}} \tag{8}$$

$$\Delta y_t = \frac{Y_{2,t} - Y_{1,t}}{Y_{2,t}} \tag{9}$$

In the above equations, $X_{1,t}$ and $X_{2,t}$ represent the 2-yr and 200-yr design values of rainfall, respectively, and $Y_{1,t}$ and $Y_{2,t}$ represent the 2-yr and 200-yr design values of tide levels, respectively. The design values of rainfall and tide levels are obtained through the joint distribution calculated in Section 3.2.

When calculating the flood change, $\Delta V_{x,t}$, due to the rainfall change in Eq. (6), there are multiple choices for tide levels. We calculate the flood change $\Delta V x_{1,t}$ caused by the tide level of 2-yr return period and rainfall changing from 2-yr return period to 200-yr return period, as shown in Eq. (10). We also calculate the flood change $\Delta V x_{2,t}$ caused by the tide level of 200-yr return period and rainfall changing from 2-yr return period to 200-yr return period, as shown in Eq. (11). The average of these two values is taken as the flood change $\Delta V_{x,t}$ due to rainfall change, as shown in Eq. (12).

$$\Delta V x_{1,t} = \frac{\left| V x_{21,t} - V x_{11,t} \right|}{\max\left( V x_{21,t}, V x_{11,t} \right)} \tag{10}$$

$$\Delta V x_{2,t} = \frac{\left| V x_{22,t} - V x_{12,t} \right|}{\max\left( V x_{22,t}, V x_{12,t} \right)} \tag{11}$$

$$\Delta V x_t = \frac{\Delta V x_{2,t} + \Delta V x_{1,t}}{2} \tag{12}$$

In the above equations, $V x_{11,t}$ represents the flood volume with the tide level of 2-yr return period and rainfall of 2-yr return period, $V x_{21,t}$ represents the flood volume with the tide level of 2-yr return period and rainfall of 200-yr return period, $V x_{12,t}$ represents the flood volume with the tide level of 200-yr return period and rainfall of 2-yr return period, and $V x_{22,t}$ represents the flood volume with the tide level of 200-yr return period and rainfall of 200-yr return period.

Similarly, when calculating the change in flood levels due to changes in tide levels, denoted as $\Delta V_{y,t}$, in Eq. (7), there are multiple choices for rainfall. We calculate the

flood change $\Delta Vy_{1,t}$ caused by the tide level changing from 2-yr return period to 200-yr return period with rainfall fixed at 2-yr return period, as shown in Eq. (13). We also calculate the flood change $\Delta Vy_{2,t}$ caused by the tide level changing from 2-yr return period to 200-yr return period with rainfall fixed at 200-yr return period, as shown in Eq. (14). The average of these two values is taken as the flood change $\Delta V_{y,t}$ due to tide level change, as shown in Eq. (15).

$$\Delta Vy_{1,t} = \frac{\left| Vx_{12,t} - Vx_{11,t} \right|}{\max\left( Vx_{12,t}, Vx_{11,t} \right)} \tag{13}$$

$$\Delta Vy_{2,t} = \frac{\left| Vx_{22,t} - Vx_{21,t} \right|}{\max\left( Vx_{22,t}, Vx_{21,t} \right)} \tag{14}$$

$$\Delta Vy_{t} = \frac{\Delta Vy_{2,t} + \Delta Vy_{1,t}}{2} \tag{15}$$

**(4) Regarding the measurement methods for rainfall and tide level data**

The rainfall and tide level data used in this study are gauge data. Rainfall data are sourced from three gauges, while tide data are from one gauge. Relevant content has been supplemented, as follows:

Both rainfall and tide level data are gauge data. There are three rainfall gauges and one tide level gauge, as shown in Fig. 1. It is assumed that rainfall distribution in the study area is uniform, and the rainfall data is averaged from the three gauges. The tide level data reflects the combined effects of astronomical tides, storm surges, and river flows. During Typhoon Mangkhut in 2018, the tide level gauge in this study recorded the highest tide level (3.28 m). (lines 104-106)

Response to the editor's comment "In addition, the reviewers find the paper lengthy and have provided several recommendations on how the manuscript can be shortened and where information that is widely available elsewhere in the literature (e.g., common equations) can be removed to streamline the paper.":

Thank you very much for your valuable suggestion. We have revised the methods and the results based on your suggestions. The specific adjustments are as follows:

(1) The formulas for correlation and joint distribution functions are provided through citations and are no longer explained in detail in the main text. The revised content is in Section 3.2.

(2) Some content less relevant to the study theme, such as schematic diagrams of different return periods, methods for designing tide level processes, model construction results, and fitting results of marginal and joint distributions, have been moved to Supplementary material.

(3) The overall language has been streamlined to make the presentation more concise.

After these revisions, the number of figures in the main text has been reduced from 15 to 11, the number of tables from 5 to 2, and the number of equations from 45 to 20. The revised manuscript presents the research methods and results more clearly and enhances the overall reading experience.

**Response to Reviewer 1**

In this study, the authors conducted a comprehensive investigation of the driving factors behind compound flooding in coastal cities using a combination of hydrodynamic modeling and mathematical statistics. Their research yielded insights into the impact of rainfall and tidal levels on compound flooding, as well as the contributions of different types of floods to compound flooding. The conclusions of this study are novel and have positive implications for risk management. The chosen topic also aligns with the scope of the journal "Hydrology and Earth System Sciences.", I have several suggestions and recommendations that would help enhance the manuscript.

**Specific comments**

1. It is necessary to indicate whether there are tide prevention facilities in the study area and how this factor is considered in the model.

**Response:** Thank you very much for your valuable question. A tidal sluice gate has been installed at the river outlet in the study area. In constructing the one-dimensional river model, we fully considered the presence of the gate. Based on your suggestion, we have supplemented the relevant information in the study area and model construction sections as follows:

(1) A tidal sluice gate, 24 m wide, has been constructed at the mouth of the Shahe River. (line 96)

(2) The tidal sluice gate is set to use tide levels as boundary conditions for outflow, ensuring that the model accounts for tide effects. The gate is opened when the water level inside the gate is higher than the level outside. The gate status is checked every hour and adjusted if necessary. (lines 195-197)

2. In the "Model construction and validation" section, a one-dimensional and two-dimensional hydrodynamic coupling model was constructed, but the analysis of flood severity primarily relied on flood volume, with limited analysis on indicators such as flood area and flood depth. This aspect needs to be supplemented.

**Response:** Thank you very much for your valuable suggestion. The severity of flooding is typically characterized by indicators such as flood volume, area, and depth. Flood area is used to determine the extent of flood impact, while flood depth reflects the flood risk level. With increasing return periods, changes in flood area are relatively small,

whereas changes in flood depth are more significant. Therefore, using flood area or flood depth alone is not suitable for calculating the impact index. Within a certain area, flood volume is the product of flood area and flood depth, providing a more comprehensive measure. Thus, in this study, flood volume is used to characterize flood severity when calculating the impact index. We have supplemented this in the methods section:

To quantify the relative contributions of rainfall and tide levels to compound flooding, this study introduces the impact index of rainfall on flooding and the impact index of tide levels on flooding. The severity of flooding is typically characterized by indicators such as flood volume, flood area, and flood depth. Within a given area, flood volume, as the product of flood area and flood depth, provides a more comprehensive measure. Therefore, in calculating the impact index, flood volume is used to represent flood severity. (lines 212-216)

Additionally, in Section 4.4 of the revised manuscript, we analyzed the spatial characteristics of compound flooding, including discussions on flood area and flood depth. The results indicate that with increasing duration and return period, compound flooding not only becomes more severe but also the severe areas gradually concentrate downstream. The contents are as follows:

From the results of maximum inundation depth in Fig. 9, it is evident that flood risk significantly increases with longer rainfall durations and higher return periods. Specifically, the flood extent expands in DU1, DU2, DU4, DU5, DU10, and DU13, with flood risk levels rising from moderate (0.5~1 m) to severe (more than 1 m). Furthermore, there is severe waterlogging in DU8, DU14 to DU22, and moderate waterlogging in DU4 also increase. It is important to note that the most severe flood risk area is located midstream, particularly under a 1-h duration and a 2-yr return period. However, under a 24-h duration and a 200-yr return period, the most severe area shifts downstream. Combined with the analysis in Section 4.2, this is because the influence of tide levels on flooding gradually strengthens with increasing rainfall duration, particularly affecting downstream areas. This indicates that tide levels drive the shift of compound flood risk towards downstream areas in coastal cities. (lines 398-405)

3. The paper only mentioned the length of the original data but did not elaborate on the sample data situation for constructing the Copula function.

**Response:** Thank you very much for your valuable question. We used the threshold method to select rainfall samples and took the highest tide level value of the same day

as the tide level samples. The sample sizes for durations of 1 h, 3 h, 6 h, 12 h, and 24 h are 48, 39, 49, 49, and 52, respectively. The specific content has been supplemented in the methods section:

This study selects samples for joint distribution from rainfall and tide level data spanning 2006 to 2021. First, the annual maximum rainfall is calculated each year to form set $D$, with the minimum value in set $D$ serving as the sample threshold. Subsequently, events from the 16-year historical data that are greater than or equal to this threshold are selected as preliminary rainfall samples. During sample selection, if the dry period between two samples is less than 6 hours, the sample with the larger rainfall is chosen. Additionally, for multiple events exceeding the threshold on the same day, only the event with the maximum rainfall is retained, resulting in the final rainfall sample. For each selected rainfall sample, the highest tide level value on the same day is identified as the tide level sample. The rainfall thresholds corresponding to durations of 1 h, 3 h, 6 h, 12 h, and 24 h are 36 mm, 56 mm, 58 mm, 66 mm, and 78 mm, respectively. The final sample sizes are 48, 39, 49, 49, and 52, respectively. (lines 135-143)

4. The paper is lengthy, and to be concise, compressing the research methods section would be helpful. For example, common formulas in Correlation and Copula.

**Response:** Thank you very much for your valuable suggestion. We have revised the methods and the results based on your suggestions. The specific adjustments are as follows:

(1) The formulas for correlation and joint distribution functions are provided through citations and are no longer explained in detail in the main text.

(2) Some content less relevant to the study theme, such as schematic diagrams of different return periods, methods for designing tide level processes, model construction results, and fitting results of marginal and joint distributions, have been moved to supplementary materials.

(3) The overall language has been streamlined to make the presentation more concise.

After these revisions, the number of figures in the main text has been reduced from 15 to 11, the number of tables from 5 to 2, and the number of equations from 45 to 20. The revised manuscript presents the research methods and results more clearly and

enhances the overall reading experience.

5. In the section "Spatial interaction of drainage units," the analysis of the interaction forces between different drainage units is not highly relevant to the main theme of this paper.

**Response:** Thank you very much for your careful review. The main objective of this study is to analyze the driving factors of compound flooding. It is hoped that the research findings can provide a scientific basis for flood management. The spatial interaction of drainage units can identify the drainage units most closely connected with others, thereby providing a reference for determining key areas for flood management. Relevant explanations have been supplemented in the methods section:

Flooding in each drainage unit is influenced through various pathways, including pipes, rivers, and surface routes. Identifying key drainage units most closely linked with others is crucial for effectively formulating flood management strategies. This study draws on the principles of the urban gravity model to quantify the interaction force between drainage units (Zhao et al., 2021). Spatial interaction force represents the connection strength between drainage units. A higher interaction value indicates a stronger spatial influence of the drainage unit, warranting more attention in flood management. (lines 292-296)

To make the research results more comprehensive, we have moved this part to the Section 4.4, "Spatial analysis of compound flooding and its driving factors". The revised content is as follows:

The interaction strength between different drainage units is shown in Fig. 10. On the right bank of the river (Fig. 10a), DU12 exhibits significant interaction with other units, indicating that as an intermediate node, it has notable interactions with upstream and downstream water flows. On the left bank of the river (Fig. 10b), DU22 has a strong spatial interaction with other drainage units, even with the distant DU6. This highlights that DU22 and DU12 play crucial roles in the entire drainage network and should receive focused attention in flood management. (lines 408-412)

6. In section 4.4 "Causes and prevention measures of floods in drainage units," flood prevention measures should not be discussed in the research results. It is suggested to elaborate on them in the discussion section.

**Response:** Thank you very much for your suggestion. We have followed your advice

and moved the relevant content to Section 5.3, "Implications for flood management in coastal cities," in the discussion part, and made the corresponding modifications:

The study results highlight the complexity of driving factors for compound flooding. Therefore, appropriate management measures should be adopted for different flood causes. As shown in Fig. 11, drainage units classified as Class III to V are impacted by tide levels, necessitating the use of high-capacity pumping facilities to reduce flood risk. For units classified as Class II, IV, and V, there are concerns regarding unreasonable drainage outfall elevations, with Class V also experiencing river overtopping. It is suggested to research river and outfall design standards that are more suitable for current and future climate conditions, while exploring ecological engineering methods to enhance the performance of rivers and drainage outfalls. For drainage outfalls directly affected by tidal jacking, planning external barriers to prevent tidal backflow is necessary. Additionally, the stormwater retention capacity within the city should be improved, implementing a combined retention and drainage strategy to minimize peak flood discharge, thereby alleviating the pressure from interactions among different types of flooding. This is beneficial for addressing various flood causes. Furthermore, drainage units with significant spatial interaction should receive focused attention to enhance the flood control capacity of the entire drainage system.

7. The conclusion needs further refinement.

**Response:** Thank you for your valuable suggestion. We have rewritten conclusions in Section 6 focusing on the theme of driving factors for compound flooding. The specific content is as follows:

Exploring the driving factors of compound flooding in coastal cities is of significant importance for scientific research and flood management. This study employs a hydrodynamic model and multivariate statistical methods to quantify the relative contributions of rainfall and tide levels to compound flooding in coastal cities and analyzes the interactions among different types of flooding. The driving factors of compound flooding were ultimately identified. The main conclusions are as follows:

(1) Compound flooding in coastal cities results from the combined effects of multiple factors, including rainfall, tide levels, drainage pipes, drainage outfalls, and rivers. Rainfall generates runoff that enters the drainage pipes, leading to pluvial flooding when the drainage capacity is insufficient. The pipes discharge water into rivers through drainage outfalls, but tide levels obstruct river drainage, causing fluvial flooding in areas with lower riverbank elevations.

(2) The combination of the copula function and Kendall return period method is effective for designing hydrological variable combinations. The study shows that the optimal joint distribution function for 24-h rainfall and tide levels is the Frank Copula, with an R² of 0.97 compared to the empirical distribution function. The Kendall return period lies between the "Or" return period and the "And" return period, avoiding excessively large or small combination design values.

(3) Compound flooding in coastal cities is influenced by the combined effects of rainfall and tide levels, with rainfall having a relatively greater contribution. From 1-h to 24-h durations, the impact index of rainfall and tide levels on compound flooding changes from 0.83 to 0.69 and from 0.17 to 0.31, respectively. This indicates that rainfall predominantly contributes to compound flooding, while the effect of tide levels is most significant at the 24-h duration.

(4) Rivers worsen compound flooding in coastal cities by elevating drainage outfalls and causing fluvial flooding. Due to tide levels, rivers elevate drainage outfalls for a duration up to 23.92 h, resulting in a maximum increase of 19.08% in pluvial flooding volume. More critically, fluvial flooding occurs, with its volume accounting for a maximum of 26.51% of the total flood volume.

**Technical comments**

1. L11: The word "Currently" is repeated.

**Response:** Thank you very much for your meticulous review. We have removed the repeated "Currently" in the revised manuscript.

2. L31: Change "in this year" to a specific year.

**Response:** Thank you very much for your meticulous review. Following your suggestion, we have changed "in this year" to "in 2023."

3. Figure 1: Mark all drainage outlets in the figure.

**Response:** Thank you very much for your valuable suggestion. We have supplemented the results of the model construction in the revised manuscript, as shown in Figure S2. This figure displays the locations of all drainage outfalls.

4. Table 1: Explain why the RMSE of the edge distribution function corresponding to the optimal tidal level for 3h is not the best.

**Response:** Thank you very much for your valuable question. For the 3-hour duration, although the marginal distribution function of the optimal tide level did not perform ideally in RMSE evaluation, the difference from the minimum value was small. This phenomenon is common when using multiple evaluation metrics. Therefore, we prioritized AIC as the main evaluation criterion in this study. Relevant explanations have been added to the methods section:

The fitting effectiveness is evaluated using the Akaike Information Criterion (AIC) method as the main criterion, along with the Root Mean Square Error (RMSE) and the Bayesian Information Criterion (BIC) method (Zhang and Singh, 2006). The Kolmogorov-Smirnov (K-S) test is used to verify the reasonableness of the results (Kavianpour et al., 2018). (lines 151-153)

5. L430-431: What is the specific relationship? It needs to be clarified.

**Response:** Thank you very much for your valuable question. The analysis of flood volume changes in Figure 9 (Figure S5 in revised manuscript) has been discussed in detail in Section 5.2 of the revised manuscript. The study found that, in the absence of tide level influence, for the same rainfall pattern, a larger total rainfall results in a more severe flood volume. However, different durations have different rainfall patterns and average rainfall intensities, and when the impact of average rainfall intensity is greater, the flood volume for longer durations may be smaller than for shorter durations. This indicates that flood intensity is not solely determined by total rainfall or rainfall intensity. The impact of tide levels on tidal sluice gate opening times significantly alters the changes in river flooding, thereby affecting the changes in compound flood volume. The specific content is as follows:

[revised manuscript text omitted]

6. L457-459: Drainage unit 14 appears in both cases simultaneously, please verify.

**Response:** Thank you very much for your meticulous review. We have carefully checked and confirmed that DU14 is almost unaffected by tide levels. Relevant content has been modified in the revised manuscript:

before modification:

This study quantifies the impact of rainfall and tide on flooding by $D_t$. The results of $D_t$ are presented in Fig. 12. DU1 to DU12 in Fig. 12(a) and DU13, DU14 and DU17 in Fig. 12(b) have $D_t$ values close to 1, indicating that drainage units far from the Pearl River are hardly affected by tides. For DU14 to DU16 and DU18 to DU22 in Fig. 12(b), $D_t$ values are around 0.5, which indicates that flooding is the result of the combined effects of rainfall and tides.

after modification:

The impact indices of rainfall on flooding are shown in Fig. 6. It can be seen that the impact indices for DU1 to DU14 and DU17 are close to 1, indicating that the contribution of tide levels to flooding in these drainage units is essentially zero. The impact indices for DU15 to DU16 and DU18 to DU22 are around 0.5, suggesting that flooding in these drainage units results from the combined effects of rainfall and tide levels. (lines 344-347)

7. L460-461: Same as above.

**Response:** Thank you very much for your meticulous review. Relevant content has been modified in the revised manuscript:

before modification:

In spatial terms, the $D_t$ values for DU14 to DU16 and DU18 are greater than those for DU19 to DU22, indicating that as the distance between the drainage units and the Shahe River and Pearl River increases, the impact of tides diminishes.

after modification:

The impact indices for DU15, DU16, and DU18 are higher than that for DU19 to DU22, indicating that as the distance from the drainage units to the Shahe River and Pearl River increases, the relative contribution of tide levels to compound flooding decreases. (lines 347-349)

8. L508-509: This is an observation, not a conclusion.

**Response:** Thank you very much for your meticulous review. The analysis of flood volume changes in Figure 14 (Figure 8 in the revised manuscript) has been moved to Section 5.2 of the discussion. Due to content modifications, this sentence no longer exists in the revised manuscript. Additionally, we have carefully checked the results section to ensure accurate descriptions of the phenomena.

9. Figure 14: The discussion of the causes of sudden changes is crucial.

**Response:** Thank you very much for your meticulous review. The study found that considering the effect of tide levels, the tidal sluice gate opening time changed, and this change coincided with the location of abrupt changes in flood volume. We have discussed the reasons for these abrupt changes in detail in Section 5.2, as follows:

Considering the effect of tide levels, it is observed that flood volumes for lower return periods exceed those for higher return periods under the same duration (Fig. S5). For instance, at the 3-h duration, return periods of 10 years and 20 years show a significant difference. This difference is mainly evident in the stage FP_S3 (Fig. 8). Combining the tidal sluice gate (hereinafter referred to as 'gate') opening times with flood processes (Table S4 and Fig. S6), it is found that this is because the gate opening time for a 10-yr return period is later than for a 20-yr return period. With more rainfall for the 20-yr return period, river water levels rise quickly due to increased rainfall, leading to gate opening conditions occurring earlier. In contrast, the 10-yr return period experiences less rainfall and fewer instances of pipe overflow (Fig. S6a), as the river water level rises slowly, the gate opening is delayed, ultimately leading to severe overflow as shown in Fig. S6b. Similarly, for durations of 6 h and 12 h (Table S4 and Fig. S6), the gate opening times are earlier for return periods of 20 years and 5 years respectively, resulting in smaller flood volumes than for return periods of 10 years and 3 years. This also explains the abrupt changes in FP and F_S3 in Fig. 8. The above analysis indicates that the combined effect of rainfall and tide levels does not increase proportionally with the return period. (lines 454-465)

10. L520-522: This should be in the Methods section.

**Response:** Thank you very much for your meticulous review. We have moved the relevant content to methods section and made corresponding modifications and improvements. The specific content is as follows:

To further explore the relative contributions of rainfall and tide levels to each stage, the method in Section 3.4 is used to calculate the impact index of rainfall and tide levels on each stage of flooding. For smaller return periods, the river water level may not exceed the elevation of the drainage outfall, and no fluvial flooding occurs. In this case, the flood volumes FW_S1 to FW_S5 are $V_t/2$, 0, 0, 0, and $V_t/2$, respectively. If the river water level exceeds the elevation of the drainage outfall but no fluvial flooding occurs, the flood volumes FW_S1 to FW_S5 are $FP\_S1_t$, $(V_t\text{-}FP\_S1_t\text{-}FP\_S5_t)/2$, 0, $(V_t\text{-}FP\_S1_t\text{-}FP\_S5_t)/2$, and $FP\_S5_t$. (lines 279-284)

**Response to Reviewer 2**

In this study, the authors thoroughly explored the factors driving compound flooding in an urban area by combining hydrodynamic modeling and multivariate statistics. Their research provided insights into the analysis of the relative contribution of rainfall and tidal levels on compound flooding and the roles of different flood scenarios. The study is fairly novel and findings are beneficial for flood risk management and applications. Further, the scope and findings of the study align with the scope of the journal "Hydrology and Earth System Sciences," and I recommend its publication but after some revisions. Below, the authors can find my suggestions and recommendations, which I believe would contribute to improving the manuscript and clarifying the methodology.

**Major Comments:**

1. Copulas and marginal distributions are fitted to the "tidal levels." Is this just the astronomical tide, or do these values include wind-driven surge (in the case of Typhoon/Cyclone conditions) and river discharge components as well? The authors mention "storm surge" as a critical factor for compound flooding several times (L346, L456, L564), but it's not clear how storm surge is taken into account when modeling "tidal levels" in this study. If the tidal range is higher and the storm surge component is relatively very small (or if only the astronomical tide is considered), fitting an extreme value model might be fundamentally incorrect since the astronomical tide is deterministic. Therefore, a clear explanation of the terminology is needed. Furthermore, are the tidal boundary conditions used at the sluice gate (as outflow boundary conditions?)? The authors mention a tidal gate, but the location is not shown in the figure.

**Response:** Thank you very much for your meticulous review and valuable suggestions. In this study, "tide level" encompasses not only astronomical tides but also water level rises caused by wind-driven surges and river flows. The tide level data used for extreme value fitting are sourced from gauge data, reflecting the influence of these multiple factors. The impact of storm surges is significant. For instance, Typhoon Mangkhut in 2018 resulted in the highest recorded tide level at the gauge, reaching 3.28 m. We have supplemented the explanation of terms and revised the description of storm surges in the paper according to your suggestions. The specific adjustments are as follows:

(1) Tide levels include not only astronomical tides but are also influenced by meteorological tides, with storm surges being an extreme form of meteorological tides. (lines 35-36)

(2) The tides of the Pearl River are influenced not only by astronomical tides but also significantly by storm surges and upstream river flood flows. (lines 94-96)

(3) The tide level data reflects the combined effects of astronomical tides, storm surges, and river flows. During Typhoon Mangkhut in 2018, the tide level gauge in this study recorded the highest tide level (3.28 m). (lines 106-107)

(4) Changed "storm surge" to "tide level". (lines 38, 53)

The tidal gate mentioned in the original manuscript refers to a sluice gate, with its specific location shown in Figure 1. In the revised manuscript, we have standardized this to "tidal sluice gate." Additionally, in the model construction, tide level boundary conditions were used at the tidal sluice gate. Relevant explanations have been supplemented in the revised manuscript:

The tidal sluice gate is set to use tide levels as boundary conditions for outflow, ensuring that the model accounts for tide effects. (lines 195-196)

2. When selecting the sample for joint probability analysis, the tidal level is chosen as the "highest tidal level of the corresponding day." This method implies that the calculated tidal design levels could occur within a 24-hour period from the rainfall event. In the numerical modeling process, a single event is used as a representative for the tidal levels process, but the timing of the rainfall peak and tidal peaks in generating design scenarios is not explained. Additionally, figure 8(a) shows the peak rainfall and peak tide almost coinciding, which is rare in reality (not the most probable 200-year event). The 3-hour rainfall duration shows the peak rainfall occurring approximately half an hour after the peak tide, whereas the 12-hour rainfall duration shows the peak rainfall occurring approximately 3 hours before the peak tide. How are the corresponding time lags between peak rainfall and peak tide determined for each scenario? Are they coming from a probabilistic basis? This timing significantly impacts the resultant flood volume and might be the main reason for the variations discussed around L430.

**Response:** Thank you very much for your valuable question. We acknowledge that the explanation of design rainfall and design tide level was insufficient, and we apologize

for this oversight. In this study, we selected the tide level from May 22, 2020, as the typical tide process based on the "Guangzhou Drainage (Rainwater) and Flood Control Comprehensive Plan (2022-2035)." The typical tide level process was scaled uniformly to design the tide level process. For a rainfall duration of 24 hours, the design tide level coincides with the start and end times of the design rainfall. For rainfall durations of 1 hour, 3 hours, 6 hours, and 12 hours, the peak of the design tide level is set at the midpoint of the rainfall duration. Relevant explanations have been added to Section 3.2.3:

For short durations (1 h and 3 h), the Pilgrim & Cordery rainfall model is used to construct the rainfall processes (Pilgrim and Cordery, 1975). For longer durations (6 h, 12 h, and 24 h), the design rainfall processes are derived using the same-frequency method, referencing the outcomes of the "Technical Report on the Compilation of Guangzhou Rainstorm Intensity Formula and Design Rainfall Patterns" for calculations. The tide level on May 22, 2020, is selected as the typical tide level process according to the "Comprehensive Planning for Drainage (Rainwater) and Waterlogging Prevention in Guangzhou (2022-2035)." The typical tide level process is modified using an equal ratio amplification method to design the tide level process. More details are provided in S1 in Supplementary material. When the rainfall duration is 24 h, the design tide level shares the same start and end times as the design rainfall. For rainfall durations of 1 h, 3 h, 6 h, and 12 h, the peak tide level is set at the midpoint of the rainfall duration.

The analysis around L430 examines the changes in flood volume under the combined effects of rainfall and tide level with varying rainfall durations and return periods. We have discussed the reasons in detail in Section 5.2 of the revised manuscript. Firstly, we analyzed the flood differences under rainfall-only conditions across different return periods and durations. We then explored the changes in flood volume differences after considering tide level effects. The study indicates that the time interval between peak rainfall and peak tide level affects the gate status, thereby influencing flood volume. However, a shorter time interval does not necessarily result in more severe flooding. For instance, under a 24-hour duration, the time interval between peak rainfall and peak tide level is shorter than that for a 12-hour duration. Yet, for return periods of 2 years and 3 years, the flood volume for 24 hours is less than that for 12 hours. Thus, flood volume changes are influenced by multiple aspects of rainfall and tide level characteristics. The specific content is as follows:

[revised manuscript text omitted]

In future work, we will consider adding variables such as time interval and rainfall intensity, and establish a joint distribution of multiple variables to more comprehensively reveal the impact of rainfall and tide levels on compound flooding. Related content has been supplemented in Section 5.4, "Limitations":

Although this study quantifies the relative contributions of total rainfall and peak tide level to compound flooding under different return periods and durations, it does not consider the impact of rainfall temporal distribution and the time interval between peak rainfall and peak tide level. Future research should design multivariate joint distributions that comprehensively consider rainfall temporal characteristics and time intervals to more fully reveal the influence of natural factors on compound flooding. (lines 496-500)

3. 41 and 42 are used to evaluate the spatial interactions between drainage units (DUs). The equations only rely on the masses and distances of two drainage units. To clarify the conclusions around L445-L450, it is essential to understand what the spatial interaction force represents. For example, the spatial interaction force between DU22 and DU6 (which are far away from each other) is 0.75. Does this relatively higher value suggest a higher level of interconnectedness or influence on each other, than even adjacent drainage units? Please explain. Also, it is not clear how the spatial interaction of the drainage units relates to the main objectives of the study.

**Response:** Thank you very much for your valuable question. The urban gravity model assumes that the interaction intensity between cities is proportional to their size (mass)

and inversely proportional to their distance. The spatial interaction between drainage units is similar to the interaction between cities. Spatial interaction force represents the connection strength between drainage units. The main objective of this study is to provide a scientific basis for flood management by analyzing the driving factors of compound flooding. The study of spatial interactions between basin units can identify the drainage units most closely connected with others, providing a reference for determining key areas for flood management. Relevant explanations have been supplemented in the revised manuscript:

Flooding in each drainage unit is influenced through various pathways, including pipes, rivers, and surface routes. Identifying key drainage units most closely linked with others is crucial for effectively formulating flood management strategies. This study draws on the principles of the urban gravity model to quantify the interaction force between drainage units (Zhao et al., 2021). Spatial interaction force represents the connection strength between drainage units. A higher interaction value indicates a stronger spatial influence of the drainage unit, warranting more attention in flood management. (lines 292-296)

Theoretically, the spatial interaction force between DU22 and DU6 is higher than that between adjacent drainage units, indicating a higher interconnectedness compared to neighboring drainage units. Upon careful examination of the calculation process, we found that the drainage pipeline length of DU22 is significantly greater than that of other left-bank drainage units, resulting in a greater spatial interaction force between DU22 and other drainage units, even with the more distant DU6. This may be related to simplifications in the model assumptions. In this study, the mass indicator only includes flood volume and drainage pipeline length, which may lead to an incomplete interpretation of mass. Additionally, the equal weighting of mass and distance may introduce uncertainty. Future work will include conducting sensitivity analyses to select more mass indicators and set indicator weights to reduce result uncertainty. Relevant explanations have been supplemented in the revised manuscript:

Additionally, the analysis of spatial interaction forces among drainage units is significant for flood management, but the current indicators used have certain limitations. We plan to adopt more comprehensive indicators in the future to enhance the depth and breadth of spatial analysis, better serving the formulation of flood management strategies. (lines 500-502)

4. It is necessary to provide additional details on the types of data used in the study. For instance, specify whether rainfall data is gridded or gauge data, and if gauge data is used, indicate the number of gauges involved. What are the locations of the tidal data and rainfall used? If so, do you assume a uniformly distributed rainfall over the entire catchment?

**Response:** Thank you very much for your meticulous review of the data. The rainfall and tide level data used in this study are gauge data. Rainfall data are sourced from three gauges, while tide data are from one gauge. The locations of these gauges are shown in Figure 1 of the revised manuscript. We assume that rainfall is uniformly distributed across the entire catchment area, thus using the average of the three gauges. This assumption is based on the representativeness of the gauges and the limitations in obtaining long-sequence data. Relevant content has been supplemented in the data collection section, as follows:

Both rainfall and tide level data are gauge data. There are three rainfall gauges and one tide level gauge, as shown in Fig. 1. It is assumed that rainfall distribution in the study area is uniform, and the rainfall data is averaged from the three gauges. (lines 104-106)

5. The paper lacks a proper discussion about the main assumptions and limitations of the process. For instance, using only 16 years of data to estimate a 200-year design event introduces significant uncertainty in modeling the dependence structure and the tail distributions. Since the study proposes a "universal" method, it is crucial for readers to understand these assumptions and uncertainties, especially when the method is to be applied to other study sites.

**Response:** Thank you very much for your valuable suggestion. We have added the limitations of the study in Section 5.4 according to your suggestion. The specific content is as follows:

The data length affects not only copula modeling merely through the bivariate behavior, but it may also have an adverse effect on the marginal (Tong et al., 2015). Intuitively, longer data provide better modeling results. This study uses 16 years of data to estimate a 200-yr return period event, introducing significant uncertainty in modeling dependence structures and tail distributions. With further data accumulation, future studies are expected to utilize longer-term data to improve analysis precision and accuracy. Although this study quantifies the relative contributions of total rainfall and peak tide level to compound flooding under different return periods and durations, it

does not consider the impact of rainfall temporal distribution and the time interval between peak rainfall and peak tide level. Future research should design multivariate joint distributions that comprehensively consider rainfall temporal characteristics and time intervals to more fully reveal the influence of natural factors on compound flooding. Additionally, the analysis of spatial interaction forces among drainage units is significant for flood management, but the current indicators used have certain limitations. We plan to adopt more comprehensive indicators in the future to enhance the depth and breadth of spatial analysis, better serving the formulation of flood management strategies.

**Minor Comments**

1. L11 "currently" repeats.

**Response:** Thank you very much for your meticulous review. We have removed the repeated "Currently" in the revised manuscript.

2. In the abstract, the authors mention the Kendall return period for the combined event of rainfall and tidal levels, greater than the "Or" return period and less than the "And" return period. The Kendall return period is typically expected to fall between the "Or" and "And" scenarios for combined events. It's not a significant finding of the study to be mentioned in the abstract.

**Response:** Thank you very much for your valuable suggestion. Following your suggestion, we have removed the description of comparisons between different return period types from the abstract and have revised and refined the abstract.

before modification:

The results show that when the return periods of rainfall and tide level are both 10 years, the Kendall return period for the combined event of rainfall and tide level is 36.35 years, greater than the "Or" return period (5.40 years) and less than the "And" return period (66.88 years).

after modification:

Taking the Shahe River basin in Guangzhou, China as a case study, the results show that the combination of the copula function and Kendall return period method is effective for designing hydrological variable combinations. (lines 15-17)

3. L32, and L36, could you provide references?

**Response:** Thank you very much for your valuable suggestion. According to your suggestion, we have supplemented the references in the revised manuscript to improve the accuracy and reliability of the paper. The specific modifications are as follows:

For example, in 2023, severe rainstorms in February led to significant casualties in the state of Sao Paulo, Brazil (Marengo et al., 2024). In May, the Emilia-Romagna region in northern Italy was hit by heavy rainfall, resulting in at least 14 fatalities and 305 landslides (Dorrington et al., 2024). In July, both Pakistan and China experienced severe rainstorm and flood disasters (Jiao et al., 2024). (lines 29-32)

References are listed uniformly at the end of the document.

4. Figure 2: what represents the purple arrow from correlation analysis to "effects of social factors on flood"?

**Response:** Thank you very much for your meticulous review. We acknowledge the error in the arrow direction and apologize for it. The original intent was to convey that the analysis of "Effects of social factors on flooding" is based on "Correlation between rainfall and tide levels". We have made the necessary corrections to Figure 2 to ensure the arrows point correctly.

5. L61, in underdeveloped areas, fluvial flooding may indeed be more prevalent due to the natural terrain and lack of infrastructure to mitigate such events. However, it is not accurate to say that "only" fluvial flooding exists in these areas. Could you explain or provide references?

**Response:** Thank you very much for your valuable suggestion. We recognize that the original manuscript's statement "In undeveloped areas, only fluvial flooding exists" was inaccurate. We have revised the relevant content in the revised manuscript according to your suggestion, as follows:

In underdeveloped areas, fluvial flooding is more common due to natural topography and a lack of infrastructure. With the development of urbanization, the increase in impervious surfaces has intensified the drainage pressure, making pluvial flooding more prominent. (lines 60-62)

6. Provide references for the "Pilgrim & Cordery rainfall model", "Maximum Possible Weighting Function" and "Pilgrim & Cordery rainfall model"

**Response:** Thank you very much for your valuable suggestion. The correct expression of "Maximum Possible Weighting Function" is "Maximum likelihood method". Based on your suggestion, we have supplemented references for the " Maximum likelihood method" and the "Pilgrim & Cordery rainfall model" in the revised manuscript. The specific modifications are as follows:

For short durations (1 h and 3 h), the Pilgrim & Cordery rainfall model is used to construct the rainfall processes (Pilgrim and Cordery, 1975). (lines 167-168)

Both marginal and joint distributions are estimated using the Maximum likelihood method to estimate their parameters (Gibson et al., 2005). (lines 149-151)

References are listed uniformly at the end of the document.

7. Figure 6 seems incorrect. The theoretical distributions (GEV, Gamma) should be smooth unless you don't change the distribution parameters.

**Response:** Thank you very much for pointing out the issue in Figure 6, and we apologize for this oversight. Upon careful examination, we found that the legends for the empirical distribution curve and the theoretical distribution curve were reversed, causing confusion. We have corrected Figure 6 to ensure that all legends accurately reflect their corresponding distribution types. The theoretical distribution curve is now smooth and presented in Figure 4 of the revised manuscript.

8. What types of information does Figure 7 (b) provide? What are the insights we can gain into joint distribution probabilities? The color scale is also not given.

**Response:** Thank you very much for your detailed review. Figure 7(b) provides the joint probability of two variables under any combination. From Figure 7(b), we can observe the joint probability of two variables under any combination. It can also be seen that the joint probability less than 0.5 occupies a large area, indicating that the joint probability of rainfall and tide level exhibits a sharp peak and heavy tail characteristic, meaning that a large amount of data is concentrated in a certain interval, while other data are widely distributed across various intervals, covering a broad range. We have supplemented the relevant analysis in the revised manuscript, as follows:

From the two-dimensional joint probability distribution of rainfall and tide levels for 24-h duration (Fig. 4d), it is observed that the joint probability less than 0.5 occupies a large area, indicating that the joint probability of rainfall and tide level exhibits a sharp peak and heavy tail characteristic. This means that a large amount of data is

concentrated in a certain interval, while other data are widely distributed across various intervals, covering a broad range. (lines 317-321)

The color scale is a key factor in understanding the data distribution in the figure. We apologize for not providing complete information. We have revised Figure 7, corresponding to Figure 4d in the revised manuscript.

9. Although the peak tidal levels could be negative (depending on the datum used), the gamma distribution is selected to fit the peak tidal levels for some rainfall durations. The gamma distribution is lower bounded by zero thus making it impossible to sample negative realizations from the fitted distribution (if necessary).

**Response:** Thank you very much for your meticulous review. The lower bound of the gamma distribution is zero, making it unsuitable for fitting data that may include negative values. In our study, the peak tidal levels are all positive, based on the reference datum we selected. Therefore, using the gamma distribution in this context is appropriate. We have supplemented the revised manuscript with relevant explanations to help readers better understand the conditions for using the gamma distribution. The specific content is as follows:

First, the Generalized Extreme Value (GEV) distribution, Normal (Norm) distribution, Gamma distribution, and Weibull distribution are applied to estimate the marginal distributions for rainfall and tide level separately. It should be noted that the lower limit of the Gamma distribution is 0. (lines 144-147)

10. The extreme sample selection is not well explained. Does it consider annual maxima? How to ensure the events are independent? What is the sample size?

**Response:** Thank you very much for your attention and suggestions regarding the sample selection method. In this study, we considered the annual maximum values. First, we calculated the maximum rainfall for each year, forming set D, and used the minimum value in set D as the sample threshold. Then, from 16 years of historical data, we selected events greater than or equal to this threshold as preliminary rainfall samples. Following the principle of the minimum inter-event time of 6 hours between two samples, if the minimum inter-event time was less than 6 hours, we selected the larger of the two samples. Additionally, if multiple events within a single day exceeded the threshold, we chose only the largest one to generate the final rainfall samples. For each selected rainfall sample P, we identified the highest tide level value Z on the same day as the tide sample. In this study, the rainfall thresholds for 1-hour, 3-hour, 6-hour, 12-

hour, and 24-hour durations were 36 mm, 56 mm, 58 mm, 66 mm, and 78 mm, respectively. The final number of samples for each duration was 48, 39, 49, 49, and 52, respectively. Relevant explanations have been supplemented in the methods section:

This study selects samples for joint distribution from rainfall and tide level data spanning 2006 to 2021. First, the annual maximum rainfall is calculated each year to form set $D$, with the minimum value in set $D$ serving as the sample threshold. Subsequently, events from the 16-year historical data that are greater than or equal to this threshold are selected as preliminary rainfall samples. During sample selection, if the dry period between two samples is less than 6 hours, the sample with the larger rainfall is chosen. Additionally, for multiple events exceeding the threshold on the same day, only the event with the maximum rainfall is retained, resulting in the final rainfall sample. For each selected rainfall sample, the highest tide level value on the same day is identified as the tide level sample. The rainfall thresholds corresponding to durations of 1 h, 3 h, 6 h, 12 h, and 24 h are 36 mm, 56 mm, 58 mm, 66 mm, and 78 mm, respectively. The final sample sizes are 48, 39, 49, 49, and 52, respectively. (lines 135-143)

11. Figure 10: The study area is in the legend (red). But cannot be seen in the figure.

**Response:** Thank you very much for your meticulous review. We have revised Figure 10 to ensure that the color of the study area in the legend matches the color displayed in the figure. Figure 10 corresponds to Figure 9 in the revised manuscript.

12. Figure 12: Do these different colors represent any information? If not use a single color.

**Response:** Thank you very much for your valuable suggestion. The different colors in Figure 12 did not convey necessary information. We have modified it to a single color according to your suggestion, making the graphical representation more concise. Figure 12 corresponds to Figure 6 in the revised manuscript.

13. It's not clear how this observation is made "Despite differences in rainfall peak timing, rainfall, and tide peak timings leading to variations in calculated flooding volumes, the impact degree index is not affected by these differences"?

**Response:** Thank you very much for your meticulous review. Regarding your observation that "Despite differences in rainfall peak timing, rainfall, and tide peak timings leading to variations in calculated flooding volumes, the impact degree index

is not affected by these differences," we recognize that there was a writing error in the original text, and we sincerely apologize for this. In Section 5.2, we discussed in detail the impact of rainfall and tide level on flood volume, finding that flood volume is not proportional to total rainfall and time interval between peak rainfall and peak tide level. A more comprehensive impact degree index will be considered in future research. Relevant explanations have been supplemented in the limitations section, as follows:

Although this study quantifies the relative contributions of total rainfall and peak tide level to compound flooding under different return periods and durations, it does not consider the impact of rainfall temporal distribution and the time interval between peak rainfall and peak tide level. Future research should design multivariate joint distributions that comprehensively consider rainfall temporal characteristics and time intervals to more fully reveal the influence of natural factors on compound flooding. (lines 496-500)

14. It's not clear how this conclusion is made "This study also highlights that under the same recurrence interval, rainfall events with larger peak timings are more destructive than those with earlier peak timings"? Have you run many events for the same return period by only changing the peak timing?

**Response:** Thank you for your careful review. Regarding your mention: "This study also highlights that under the same recurrence interval, rainfall events with larger peak timings are more destructive than those with earlier peak timings," and its previous sentence "Similar conclusions have been drawn by Cheng et al., stating that floods with shorter time lags and later peak timings intensify the impact of flooding" are all derived from the work of Cheng et al. Inaccurate citation and language description caused confusion in understanding. We have reorganized the relationship between rainfall characteristics and flooding in the revised manuscript. Since this study did not design multiple peak coefficients and peak time scenarios for the same duration, this part is no longer included in the revised manuscript. Additionally, references in the revised manuscript have been verified and correctly cited from the original sources.

15. The authors suggest a set of equations (from eq. 30 to eq. 40), but poorly explained. Consider elaborating more about the governing process of equations. Additionally, it's mentioned that "calculate the range of rainfall and tidal level design values for different durations from 2-yr RP to 200-yr RP using the following formula" (L286,), but the formula doesn't give the range of rainfall, and since it divides the range by 200yr Rainfall value. So, $\Delta Xt$ will be a dimensionless parameter that is related to the amount

of variability (range) of rainfall. Same for Tides. Same in L304, ΔVxt and ΔVyt do not quantify the variation of flooding volume. Accordingly, check Dxt and Dyt.

**Response:** Thank you for your detailed review of the equations and related calculation methods in our paper. We acknowledge that our explanation and description in this section were indeed insufficient, and we apologize for this. Regarding your mention of L286, "calculate the range of rainfall and tidal level design values for different durations from 2-yr RP to 200-yr RP using the following formula. The original intention was to express "calculating the variation range of rainfall and tide levels from 2-year to 200-year rainfall and tide levels," where the rainfall and tide levels are derived from the design values. We have corrected this statement in the Section 3.4 of the revised manuscript and re-examined ΔVxt, ΔVyt, Dxt, and Dyt to ensure that all parameters and variables are clearly and correctly defined. The specific modifications are as follows:

The impact index of rainfall on flooding ($D_{x,t}$) is calculated by comparing the impact of rainfall on flooding with the total impact of both rainfall and tide levels, as shown in Eq. (4). Similarly, the impact index of tide levels on flooding ($D_{y,t}$) is calculated by comparing the impact of tide levels on flooding with the total impact of both rainfall and tide levels, as shown in Eq. (5). In this study, the impact index of tide levels on flooding can also be calculated as $1 - D_{x,t}$. The value of $D_{x,t}$ ranges from 0 to 1, where 0 indicates that flooding is solely influenced by tide levels, and 1 indicates that flooding is solely influenced by rainfall. A larger $D_{x,t}$ value signifies a more significant impact of rainfall on flooding, with a relatively weaker influence of tide levels.

$$D_{x,t} = \frac{F_{x,t}}{F_{x,t} + F_{y,t}} \tag{4}$$

$$D_{y,t} = \frac{F_{y,t}}{F_{x,t} + F_{y,t}} = 1 - D_{x,t} \tag{5}$$

In the above equations, $t$ represents different durations. The impact of rainfall on flooding, $F_{x,t}$, refers to the flood volume change due to the rainfall change. The impact of tide levels on flooding, $F_{y,t}$, refers to the flood volume change due to the tide levels change. The calculation formulas are as follows:

$$F_{x,t} = \frac{\Delta V_{x,t}}{\Delta x_t} \tag{6}$$

$$F_{y,t} = \frac{\Delta V_{y,t}}{\Delta y_t} \tag{7}$$

In the above equations, the change in rainfall, $\Delta x_t$, is the relative change from 2-yr return period to 200-yr return period, and the change in tide levels, $\Delta y_t$, is also the relative change from 2-yr return period to 200-yr return period. The calculation formulas are as follows:

$$\Delta x_t = \frac{X_{2,t} - X_{1,t}}{X_{2,t}} \tag{8}$$

$$\Delta y_t = \frac{Y_{2,t} - Y_{1,t}}{Y_{2,t}} \tag{9}$$

In the above equations, $X_{1,t}$ and $X_{2,t}$ represent the 2-yr and 200-yr design values of rainfall, respectively, and $Y_{1,t}$ and $Y_{2,t}$ represent the 2-yr and 200-yr design values of tide levels, respectively. The design values of rainfall and tide levels are obtained through the joint distribution calculated in Section 3.2.

When calculating the flood change, $\Delta V_{x,t}$, due to the rainfall change in Eq. (6), there are multiple choices for tide levels. We calculate the flood change $\Delta Vx_{1,t}$ caused by the tide level of 2-yr return period and rainfall changing from 2-yr return period to 200-yr return period, as shown in Eq. (10). We also calculate the flood change $\Delta Vx_{2,t}$ caused by the tide level of 200-yr return period and rainfall changing from 2-yr return period to 200-yr return period, as shown in Eq. (11). The average of these two values is taken as the flood change $\Delta V_{x,t}$ due to rainfall change, as shown in Eq. (12).

$$\Delta Vx_{1,t} = \frac{\left| Vx_{21,t} - Vx_{11,t} \right|}{\max\left( Vx_{21,t}, Vx_{11,t} \right)} \tag{10}$$

$$\Delta Vx_{2,t} = \frac{\left| Vx_{22,t} - Vx_{12,t} \right|}{\max\left( Vx_{22,t}, Vx_{12,t} \right)} \tag{11}$$

$$\Delta Vx_t = \frac{\Delta Vx_{2,t} + \Delta Vx_{1,t}}{2} \tag{12}$$

In the above equations, $Vx_{11,t}$ represents the flood volume with the tide level of 2-yr return period and rainfall of 2-yr return period, $Vx_{21,t}$ represents the flood volume with the tide level of 2-yr return period and rainfall of 200-yr return period, $Vx_{12,t}$ represents the flood volume with the tide level of 200-yr return period and rainfall of 2-yr return period, and $Vx_{22,t}$ represents the flood volume with the tide level of 200-yr return period and rainfall of 200-yr return period.

Similarly, when calculating the change in flood levels due to changes in tide levels, denoted as $\Delta V_{y,t}$, in Eq. (7), there are multiple choices for rainfall. We calculate the flood change $\Delta Vy_{1,t}$ caused by the tide level changing from 2-yr return period to 200-yr return period with rainfall fixed at 2-yr return period, as shown in Eq. (13). We also calculate the flood change $\Delta Vy_{2,t}$ caused by the tide level changing from 2-yr return period to 200-yr return period with rainfall fixed at 200-yr return period, as shown in Eq. (14). The average of these two values is taken as the flood change $\Delta V_{y,t}$ due to tide level change, as shown in Eq. (15).

$$\Delta Vy_{1,t} = \frac{\left|Vx_{12,t} - Vx_{11,t}\right|}{\max\left(Vx_{12,t}, Vx_{11,t}\right)} \tag{13}$$

$$\Delta Vy_{2,t} = \frac{\left|Vx_{22,t} - Vx_{21,t}\right|}{\max\left(Vx_{22,t}, Vx_{21,t}\right)} \tag{14}$$

$$\Delta Vy_t = \frac{\Delta Vy_{2,t} + \Delta Vy_{1,t}}{2} \tag{15}$$

16. The paragraph from L543 to 558 is more about discussing the results. Consider moving it into the discussion section.

**Response:** Thank you very much for your suggestion. We have followed your advice and moved the relevant content to Section 5.3, "Implications for flood management in coastal cities," in the discussion part, and made the corresponding modifications:

The study results highlight the complexity of driving factors for compound flooding. Therefore, appropriate management measures should be adopted for different flood causes. As shown in Fig. 11, drainage units classified as Class III to V are impacted by tide levels, necessitating the use of high-capacity pumping facilities to reduce flood risk. For units classified as Class II, IV, and V, there are concerns regarding unreasonable drainage outfall elevations, with Class V also experiencing

river overtopping. It is suggested to research river and outfall design standards that are more suitable for current and future climate conditions, while exploring ecological engineering methods to enhance the performance of rivers and drainage outfalls. For drainage outfalls directly affected by tidal jacking, planning external barriers to prevent tidal backflow is necessary. Additionally, the stormwater retention capacity within the city should be improved, implementing a combined retention and drainage strategy to minimize peak flood discharge, thereby alleviating the pressure from interactions among different types of flooding. This is beneficial for addressing various flood causes. Furthermore, drainage units with significant spatial interaction should receive focused attention to enhance the flood control capacity of the entire drainage system.

17. The paper is lengthy. Consider moving some of the extra results to the supplementary materials. For example, Tables 1 and 2. Commonly used equations, such as those for correlation and distributions, can be omitted from the text. Instead, it is sufficient to cite the relevant references for these well-known equations.

**Response:** Thank you very much for your valuable suggestion. We have revised the methods and the results based on your suggestions. The specific adjustments are as follows:

(1) The formulas for correlation and joint distribution functions are provided through citations and are no longer explained in detail in the main text.

(2) Some content less relevant to the study theme, such as schematic diagrams of different return periods, methods for designing tide level processes, model construction results, and fitting results of marginal and joint distributions, have been moved to supplementary materials.

(3) The overall language has been streamlined to make the presentation more concise.

After these revisions, the number of figures in the main text has been reduced from 15 to 11, the number of tables from 5 to 2, and the number of equations from 45 to 20. The revised manuscript presents the research methods and results more clearly and enhances the overall reading experience.

18. Check the way of citing publications in the manuscript according to the HESS guideline (e.g., L40, L42, L50).

**Response:** Thank you for pointing out the issue with the citation format. We have

carefully checked and revised the reference format to ensure they strictly adhere to the guidelines of the journal "Hydrology and Earth System Sciences". The following are part of the revisions made:

(1) L40

before modification:

[revised manuscript text omitted]